# Causal PETS: Causality-Informed PET Synthesis from Multi-modal Data

**Yujia Li**[*1,2]                                                    YUJIA.LI@MIRACLE.ICT.AC.CN

[1] *Key Laboratory of Intelligent Information Processing of Chinese Academy of Sciences (CAS), Institute of Computing Technology, CAS*
[2] *University of Chinese Academy of Sciences*

**Han Li**[*3]

[3] *Computer Aided Medical Procedures (CAMP), School of Computation, Information and Technology, Technische Universitaet Muenchen (TUM)*

**S Kevin Zhou**[1,4,5,6]                                             SKEVINZHOU@USTC.EDU.CN

[4] *School of Biomedical Engineering, Division of Life Sciences and Medicine, University of Science and Technology of China (USTC), Hefei Anhui, 230026, China*
[5] *Center for Medical Imaging, Robotics, Analytic Computing & Learning (MIRACLE), Suzhou Institute for Advance Research, USTC, Suzhou Jiangsu, 215123, China*
[6] *Key Laboratory of Precision and Intelligent Chemistry, USTC, Hefei Anhui, 230026, China*

**Editors:** Accepted for publication at MIDL 2025

## Abstract

Positron emission tomography (PET) plays a crucial role in diagnosing and monitoring neurological disorders. However, its clinical availability is constrained by high costs, radiation exposure risks, and logistical limitations. In this study, we propose **Causal PETS**, a novel causality-informed multimodal synthesis model for PET image generation. Unlike conventional approaches that rely on a direct transformation from $T_1$-weighted MRI to PET, our model explicitly captures causal relationships among multimodal data—including MRI, demographic information, and cerebrospinal fluid (CSF) biomarkers—and seamlessly integrates these factors into the PET synthesis process. Through extensive evaluations, we demonstrate that Causal PETS surpasses existing non-causal methods in image clarity and accuracy, particularly in highlighting regions of interest critical for neurological disorders such as Alzheimer's disease (AD). This work underscores the significance of causality in medical image synthesis and highlights the potential of multimodal integration for enhancing clinical decision-making.

**Keywords:** medical image synthesis, causality, Alzheimer's Disease

## 1. Introduction

Medical image synthesis offers new solutions to the problem of obtaining certain modality imaging data. For example, Positron Emission Tomography (PET) is pivotal in neuroimaging and is crucial for the diagnosis and monitoring of neurodegenerative diseases such as Alzheimer's Disease (AD) by detecting abnormal molecules (Marcus et al., 2014; Nordberg et al., 2010). However, the widespread use of PET is hindered by several challenges: its reliance on frequent scans for longitudinal studies, the significant expenses of the radiotracers and advanced imaging technology, and the health risks posed by radiation exposure

---

[*] Equal contribution

(Nievelstein et al., 2012; Brix et al., 2009). Thus, there is an urgent need to explore alternative approaches for acquiring PET to support diagnostic applications, among which the synthesis of PET from other more available modalities presents a promising solution.

Recently, deep learning-based medical image synthesis models have shown great potential. (Wang et al., 2021a,b). In the context of PET imaging, most approaches adopt a straightforward architecture for generating target images from source images. Some research synthesize PET images from MRI or CT (Zhang et al., 2022a; Ou et al., 2024b), while others aim at generating high-dose PET images from low-dose PET data (Pan et al., 2024; Shen et al., 2024). While achieving success, this type of image-to-image paradigm, as shown in Fig. 1 (blue path), faces one key challenge: the significant information gap between two image modalities. For instance, generating PET-CT from MRI often suffers from insufficient information representation. The imaging principle of MRI relies on differences in proton relaxation times, which provides structural information. In contrast, PET-CT imaging, based on the positron radiation of radioactive tracers binding to specific molecules, reflects the distribution of these molecules, which MRI inherently lacks.

Multi-modal image synthesis, which integrates complementary information from other modalities, serves as an effective approach to filling the information gap between two image modalities. However, existing multi-modal image synthesis methods typically take all modalities as direct inputs and rely on the network to automatically learn how to utilize them. This approach often suffers from limitations, such as suboptimal exploitation of relationships between different modalities, which in turn leads to inefficient feature fusion and degraded synthesis performance.

To address these challenges, causal image synthesis provides a principled framework by explicitly modeling the causal relationships among different modalities. Deep Structural Causal Models (Pawlowski et al., 2020) and Neural Causal Models (Xia et al., 2022) have demonstrated the potential of incorporating causal structures into generative models. However, they exhibit several limitations. DSCMs primarily rely on VAE for image generation, which often leads to blurry and less realistic synthetic images. In addition, they focus on counterfactual image generation, which lacks ground-truth validation, making it difficult to assess the accuracy and reliability of the generated images in real-world applications. These limitations motivate our work, which extends causal image synthesis to a multi-modal, high-fidelity, and ground-truth-verifiable setting, ensuring both interpretability and practical utility.

We propose an innovative causality-informed multi-modal synthesis model that explicitly models and leverages causal relationships between multi-modalities to better exploit their complementary information, fill the information gap between image modalities, and improve synthesis performance. Specifically, the novelties and contributions of this paper are as follows: (i) **A novel causality-informed multimodal** medical image synthesis paradigm. Unlike conventional image-to-image translation methods that rely on a single modality, Causal PETS employs a causal graph to explicitly model and leverage causal relationships among multiple modalities, effectively capturing their complementary information for improved PET image synthesis. (ii) Enhanced Performance. Leveraging the causal graph, Causal PETS achieves state-of-the-art (SOTA) reconstruction quality in PET image synthesis. Furthermore, by integrating the synthesized PET images with existing modality data, our approach also attains SOTA classification performance for the early diagnosis

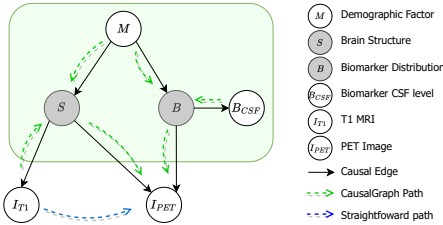

Figure 1: The Causal Graph of PET Synthesis

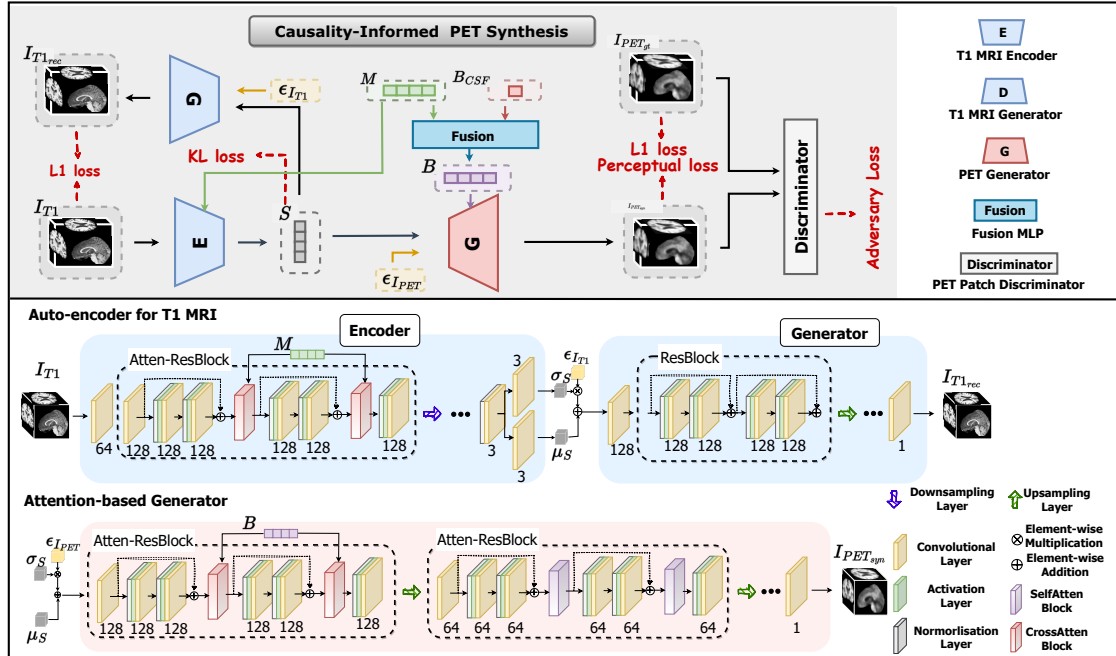

Figure 2: The architecture of the auto-encoder of $T_1$ MRI and the PET generator.

and monitoring of Alzheimer's disease (AD). (iii) **High interpretability**. By generating PET images under controlled interventions on specific variables in the model, Causal PETS enables a deeper understanding of the role of each modality, enhancing the interpretability of the synthesis process.

## 2. Method: Causal PETS

### 2.1. Causal Graph & Structural Causal Equations

#### 2.1.1. CAUSAL GRAPH

A Causal Graph (Pearl, 2010) is a graph representing the causal relationships between variables, where nodes represent variables and an edge from node A to node B (A→B) signifies that A has a direct causal effect on B. The variables can be either **Observed Variables**, measured in the dataset or **Latent Variables**, not directly observed.

We construct the causal graph of PET acquisition for AD as Fig.1, which ontains the demographic factor $M$ (Age, Gender, Education levels, APOE4 allele number), brain structure $S$, biomarker CSF level $B_{CSF}$, biomarker distribution $B$, $T_1$ MRI image $I_{T_1}$, and PET image $I_{PET}$, according to clinical research, explained as follows.

$M \rightarrow S$: Demographic factors have a causal impact on brain structure. Brain atrophy occurs as a results of age increasing (Kasantikul et al., 1979), differences in brain volumes are significant between males and females (Gur et al., 2002), and both educational levels and the presence of the APOE4 allele contribute to brain atrophy (Coffey et al., 1999; Lim et al., 2017).

$S \rightarrow I_{T1}$: $T_1$ MRI relies on the magnetic resonance signals of hydrogen nuclei(Tang et al., 2018) and is directly caused by the brain structure.

$S \rightarrow I_{PET}$: PET imaging (Catafau and Bullich, 2015) relies on the use of radiotracers the uptake of radiotracers may vary depending on the size and shape of brain ($S$).

$B \rightarrow I_{PET}$: The radiotracers bind to specific molecules and the molecule concentration distribution($B$) determines the radiotracer uptake and thus influences $I_{PET}$ (Catafau and Bullich, 2015).

$B \rightarrow B_{CSF}$: CSF biomarkers measure the sampled concentration of molecules in the cerebrospinal fluid (CSF), thus determined by the distribution of specific molecule.

### 2.1.2. Structural Causal Equations

Structural Causal Equations show how a variable is generate, which can be expressed as

$$Y = f_Y(\text{Pa}(Y), \epsilon_Y), \tag{1}$$

where $f_Y$ is a generative function, $\text{Pa}(Y)$ denotes the set of parent variables of $Y$, and $\epsilon_Y$ is an error term representing all other latent variables affecting $Y$.

Take the variable $I_{PET}$ as an example. The structural equation is expressed as

$$I_{PET} = f_{I_{PET}}(S, B, \epsilon_{I_{PET}}), \tag{2}$$

where $\epsilon_Y$ can be the PET image prototype, the instrumental or the imaging noise, which can affect the PET image but are not included in the model.

For the PET synthesis model, $I_{T1}$ and $B_{CSF}$ can be used to provide the information of their causal parents. In our model, we predict the posterior $S$ and $B$ by

$$S = g_S(I_{T_1}, M), \quad B = g_B(B_{CSF}, M), \tag{3}$$

where $g_S$ is implemented by an encoder and $g_B$ by an Multiple-Layer Perceptron (MLP). Then we use an decoder to implement $f_{I_{PET}}$ in (2).

## 2.2. Causality-Informed PET Synthesis (Causal PETS)

The Causal PETS Model is shown in Fig. 2, based on the causal graph Fig. 1. We use two decoders as the structural equation for $T_1$ MRI and PET, described in (2).

When training and infering, the $S$ and $B$ are firstly approximated by

$$S = E(I_{T_1}, M), \quad B = f(B_{CSF}, M), \tag{4}$$

where $f$ denotes the fusion MLP and $E$ denotes the T$_1$ MRI encoder.

Then $I_{PET_{syn}}$ and $I_{T1_{rec}}$ are generated as the causal structural function,

$$I_{PET_{syn}} = G_P(S, B, \epsilon_{PET}), \quad I_{T1_{rec}} = G_{T_1}(S, \epsilon_{T1}), \tag{5}$$

where $G_P$ and $G_{T_1}$ denotes generator, $\epsilon_{PET}$ and $\epsilon_{T_1}$ are sampled from normal distribution.

As for the better quality of PET synthesis, we added a discriminator $D$ for adversary training, matching the synthetic data distribution to the target data distribution.

### 2.2.1. Architectures

In this section we introduce the architectures of networks of the proposed model. Fig. 2 provides an architectural overview. The model architecture details can be found in code[1].

**Auto-encoder for T$_1$**   The auto-encoder consists of one Atten-ResBlock, five ResBlocks, three Upsample Layers, and three Downsample Layers. The encoder predict the $\mu_S$ and $\sigma_S$ and the generator outputs the $I_{T1_{rec}}$. The dimension of feature map is marked in Fig. 2.

**Attention-based Generator**   The attention-based generator consists of three Atten-ResBlocks and three upsampling layers. The first Atten-ResBlock is of cross-attention and the other two are of self-attention.

**Fusion MLP and Discriminator**   The fusion MLP is made up with three linear layers of a latent dimension 128, and the discriminator is chosen as a Patch Discriminator, a discriminator structure based on PatchGAN (Isola et al., 2017a).

### 2.2.2. Loss Function

For the auto-encoder, the commonly used reconstruction loss and the KL loss is used as

$$\mathcal{L}_{\text{AE}} = \mathbb{E}_{x \sim I_{T_1}} \left[ \|I_{T_1} - G_{T1}(E(I_{T_1}))\|^2 \right] + \text{KL}\left( q_\phi(E(I_{T_1})) \| p(S) \right), \tag{6}$$

where $\text{KL}\left( q_\phi(S|I_{T_1}) \| p(S) \right)$ is the KL divergence between the approximate posterior and the prior distribution (normal distribution)

For the PET image generator, an $L_1$ loss and a perceptual loss are incorporated to minimize the absolute pixel-wise difference and the perceptual difference.

$$\mathcal{L}_1(G_P) = \mathbb{E}_{(x,z,m,y) \sim (I_{T_1}, B_{CSF}, M, I_{PET}), \epsilon \sim \mathcal{N}} \|y - G_P(E(x, m), f(z), \epsilon)\|_1, \tag{7}$$

$$\mathcal{L}_{\text{Perceptual}}(G_P) = \mathbb{E}_{(x,z,m,y) \sim (I_{T_1}, B_{CSF}, M, I_{PET}), \epsilon \sim \mathcal{N}} \|V(y) - V(G_P(E(x, m), f(z, m), \epsilon))\|_1. \tag{8}$$

The loss function of the generator $G_P$ and the discriminator $D$ is as follows,

$$\mathcal{L}(D) = \mathbb{E}_{(x,z,m) \sim (I_{T_1}, B_{CSF}, M), \epsilon \sim \mathcal{N}} \left[ (D(G_P(E(x, m), f(z, m), \epsilon))^2 \right] + \mathbb{E}_{y \sim I_{PET}} \left[ (D(y) - 1)^2 \right], \tag{9}$$

$$\mathcal{L}_{adv}(G_P) = \mathbb{E}_{x \sim I_{T1}, z \sim (B_{CSF}, M), \epsilon \sim \mathcal{N}} [1 - (D(G_P(E(x, m), f(z), \epsilon))^2]. \tag{10}$$

---

1. https://github.com/jessyblues/Causality-Informed-PET-Synthesis-from-Multi-modal-Data

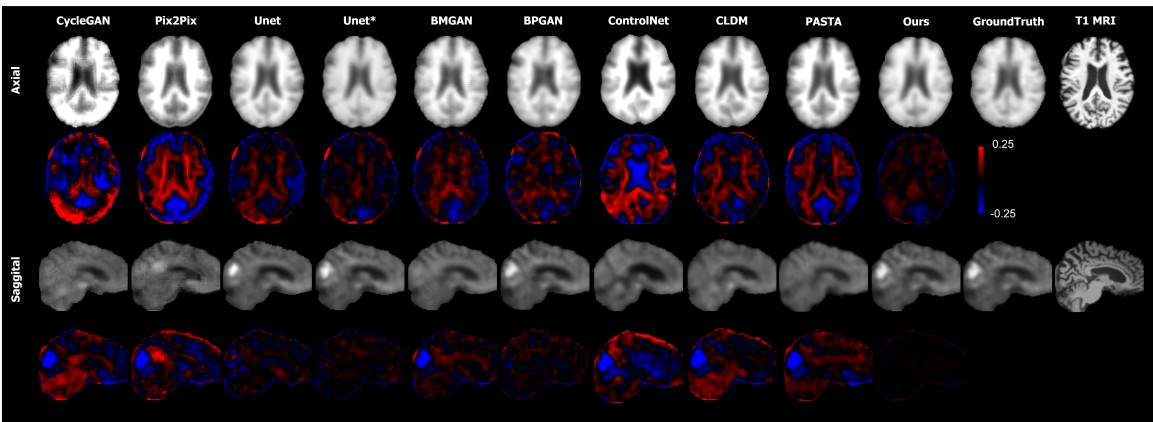

Figure 3: The visualizations of synthesized AV45 PET (the first row) and AV1451 PET (the third row) of different methods with the different map against the grountruth.

The overall loss function are as follows:

$$\mathcal{L}_{\text{AE},G_P} = \mathcal{L}_{\text{AE}} + \mathcal{L}_1(G_P) + \lambda_p \mathcal{L}_{Perceptual}(G_P) + \lambda_{adv} \mathcal{L}_{adv}(G_P). \tag{11}$$

where $\lambda_p$ and $\lambda_{adv}$ are the hyper-parameters and the details are in appendix. The discriminator and generator is trained using a standard adversarial framework.

## 3. Experiments

### 3.1. Datasets and Training Details

We train and test on a subset of the Alzheimer's Disease Neuroimaging Initiative database (ADNI) (Petersen et al., 2010), using PET of two different radio-tracers, AV45 and AV1451, with corresponding CSF bio-marker data. The details of dataset are in the appendix.

| | dataset AV45 | | | | dataset AV1451 | | | |
|---|---|---|---|---|---|---|---|---|
| Method | MAE($\times 10^{-1}$)↓ | SSIM(%)↑ | PNSR↑ | $\epsilon_{SUVR}$↓ | MAE($\times 10^{-1}$)↓ | SSIM(%)↑ | PNSR↑ | $\epsilon_{SUVR}$↓ |
| CycleGAN (Zhu et al., 2017) | $0.349^{***}_{\pm 0.147}$ | $92.46^{***}_{\pm 1.50}$ | $23.192^{***}_{\pm 2.558}$ | $0.23^{***}_{\pm 0.30}$ | $0.443^{***}_{\pm 0.219}$ | $88.26^{***}_{\pm 2.10}$ | $21.869^{***}_{\pm 3.263}$ | $0.12^{***}_{\pm 0.11}$ |
| Pix2Pix (Isola et al., 2017b) | $0.450^{***}_{\pm 0.164}$ | $93.78^{***}_{\pm 1.46}$ | $21.825^{***}_{\pm 3.001}$ | $0.21^{***}_{\pm 0.27}$ | $0.251^{***}_{\pm 0.081}$ | $91.34^{***}_{\pm 1.50}$ | $25.469^{***}_{\pm 1.830}$ | $0.12^{***}_{\pm 0.12}$ |
| U-Net w/o condition | $0.265^{***}_{\pm 0.122}$ | $96.73^{***}_{\pm 1.33}$ | $25.719^{***}_{\pm 2.930}$ | $0.16^{**}_{\pm 0.17}$ | $0.230^{***}_{\pm 0.112}$ | $95.42^{***}_{\pm 1.37}$ | $26.216^{***}_{\pm 2.610}$ | $0.11^{***}_{\pm 0.10}$ |
| U-Net w/ condition | $0.237^{**}_{\pm 0.142}$ | $97.19^{***}_{\pm 1.28}$ | $26.578^{***}_{\pm 2.859}$ | $0.12^{**}_{\pm 0.13}$ | $0.229^{***}_{\pm 0.119}$ | $95.15^{***}_{\pm 1.51}$ | $26.425^{***}_{\pm 2.668}$ | $0.09^{**}_{\pm 0.09}$ |
| BMGAN (Hu et al., 2021) | $0.320^{***}_{\pm 0.116}$ | $96.51^{***}_{\pm 1.41}$ | $24.262^{***}_{\pm 2.374}$ | $0.18^{***}_{\pm 0.22}$ | $0.214^{*}_{\pm 0.136}$ | $94.78^{***}_{\pm 1.53}$ | $27.644^{***}_{\pm 3.691}$ | $0.10^{*}_{\pm 0.11}$ |
| BPGAN (Zhang et al., 2022b) | $0.321^{***}_{\pm 0.114}$ | $95.63^{***}_{\pm 1.31}$ | $24.092^{***}_{\pm 2.215}$ | $0.18^{***}_{\pm 0.24}$ | $0.214^{*}_{\pm 0.162}$ | $93.97^{***}_{\pm 1.49}$ | $27.688^{***}_{\pm 4.285}$ | $0.11^{**}_{\pm 0.10}$ |
| ControlNet (Zhang et al., 2023) | $0.553^{***}_{\pm 0.081}$ | $91.90^{***}_{\pm 1.78}$ | $21.720^{***}_{\pm 1.107}$ | $0.17^{***}_{\pm 0.16}$ | $0.941^{***}_{\pm 0.167}$ | $81.77^{***}_{\pm 2.66}$ | $19.296^{***}_{\pm 1.415}$ | $0.10^{***}_{\pm 0.09}$ |
| CLDM (Ou et al., 2024a) | $0.329^{***}_{\pm 0.128}$ | $95.57^{***}_{\pm 1.26}$ | $23.905^{***}_{\pm 2.833}$ | $0.23^{***}_{\pm 0.15}$ | $0.211_{\pm 0.117}$ | $97.07^{***}_{\pm 2.92}$ | $27.896^{***}_{\pm 3.215}$ | $0.11^{*}_{\pm 0.12}$ |
| PASTA (Li et al., 2024) | $0.349^{***}_{\pm 0.109}$ | $95.33^{***}_{\pm 1.20}$ | $23.513^{***}_{\pm 2.393}$ | $0.23^{***}_{\pm 0.14}$ | $0.394^{***}_{\pm 0.171}$ | $93.26^{***}_{\pm 2.73}$ | $23.104^{***}_{\pm 3.155}$ | $0.12^{*}_{\pm 0.09}$ |
| **Causal PETS (ours)** | $\mathbf{0.224_{\pm 0.104}}$ | $\mathbf{97.47_{\pm 1.13}}$ | $\mathbf{26.740_{\pm 2.581}}$ | $\mathbf{0.10_{\pm 0.13}}$ | $\mathbf{0.202_{\pm 0.096}}$ | $\mathbf{98.12_{\pm 0.82}}$ | $\mathbf{29.687_{\pm 1.905}}$ | $\mathbf{0.08_{\pm 0.10}}$ |

Table 1: Quantitative comparison of PET images synthesised by different methods

### 3.2. PET Image Quality

We evaluate our model's performance in generating PET images, compared against image translation methods Pix2Pix (Isola et al., 2017b), CycleGAN (Zhu et al., 2017), Unet, and MRI-specific networks including BMGAN (Hu et al., 2021), BPGAN (Zhang et al.,

2022b) and diffusion-based method including ControlNet (Zhang et al., 2023), CLDM (Ou et al., 2024a), and PASTA (Li et al., 2024). Quantitative results, including mean absolute error (MAE), multi-sacle structural similarity (SSIM) index, and Peak Signal-to-Noise Ratio (PSNR), are detailed in Table 1. We use paired t-tests to evaluate statistical significance. Statistical significance is indicated as ***: $p < 0.001$, **: $p < 0.01$, and *: $p < 0.05$.

Specifically, we furthermore compute the SUVR (Standardized Uptake Value Ratio) MAE between the synthesized PET and the target real PET. SUVR is a metric to quantify the concentration of radio-tracer uptake in a region of interest (ROI) relative to a reference region. As recommended in clinical research (Schindler et al., 2021; Jack Jr et al., 2018), the cerebral cortex region is set as the ROI with the cerebellar cortex as the reference region. The formula of SUVR computation is provided in the appendix.

Our method achieves the lowest MAE, the highest SSIM and PSNR on both datasets, demonstrating a superior accuracy and the advanced structural similarity. Our method also outperforms other methods in terms of SUVR MAE, demonstrating its effectiveness in synthesizing high-quality PET images in terms of ROI.

Figs. 3 show the slices of the synthesised AV45 and AV1451 PET images respectively. The error map is visualized by subtracting the real PET image from the synthetic one. It shows that our method generates the most authentic PET image and the darkest error map.

### 3.3. Downstream Tasks

| Method | dataset AV45 | | | | | dataset AV1451 | | | | |
|---|---|---|---|---|---|---|---|---|---|---|
| | F1 | AUC | Acc | Prec | Recall | F1 | AUC | Acc | Prec | Recall |
| CycleGAN (Zhu et al., 2017) | 0.7494*** | 0.6148*** | 0.7857*** | 0.7221*** | 0.7857*** | 0.8616*** | 0.9245*** | 0.8958*** | 0.9069*** | 0.8558*** |
| Pix2Pix (Isola et al., 2017b) | 0.7629*** | 0.5948*** | 0.7653*** | 0.7606*** | 0.7653*** | 0.8977*** | 0.8849*** | 0.9167*** | **0.9823** | 0.8167*** |
| Unet w/o condition | 0.7747*** | 0.6016*** | 0.7951*** | 0.8032*** | 0.7551*** | 0.8977*** | 0.9202** | 0.9167*** | 0.9239*** | 0.9167*** |
| Unet w/ condition | 0.7990*** | 0.8301* | 0.8129* | 0.8137*** | 0.8129 | 0.9093*** | 0.9405* | 0.9167*** | 0.9091*** | 0.9167*** |
| BMGAN (Hu et al., 2021) | 0.7337*** | 0.5478*** | 0.7163*** | 0.6663*** | **0.8163** | 0.8425*** | 0.9446 | 0.8021*** | 0.9271*** | 0.8021*** |
| BPGAN (Zhang et al., 2022b) | 0.7520*** | 0.6781*** | 0.8095** | 0.7461*** | 0.7795*** | 0.8977*** | 0.9302** | 0.9167*** | 0.9239*** | 0.9167*** |
| ControlNet (Zhang et al., 2023) | 0.6620*** | 0.7225*** | 0.7333*** | 0.557*** | 0.6333*** | 0.7038*** | 0.7124*** | 0.4746*** | 0.5458*** | 0.6458*** |
| CLDM (Ou et al., 2024a) | 0.6320*** | 0.7584*** | 0.8036*** | 0.6526*** | 0.6199*** | 0.8824*** | 0.9384** | 0.8750*** | 0.8937*** | 0.8750*** |
| PASTA (Li et al., 2024) | 0.4156*** | 0.7500*** | 0.5714*** | 0.3265*** | 0.5714*** | 0.8167*** | 0.9483 | 0.8750*** | 0.7656*** | 0.8750*** |
| **Causal PETS (ours)** | **0.8310** | **0.8373** | **0.8265** | **0.8782** | 0.8087* | **0.9547** | **0.9505** | **0.9583** | 0.9602*** | **0.9583** |
| Real Images | 0.8587 | 0.8569 | 0.8265 | 0.9178 | 0.8465 | 0.9592 | 0.9898 | 0.9615 | 0.9632 | 0.9615 |

Table 2: Comparison of pMCI vs sMCI classification results using synthesised PET images.

| Method | dataset AV45 | | | | | dataset AV1451 | | | | |
|---|---|---|---|---|---|---|---|---|---|---|
| | F1 | AUC | Acc | Prec | Recall | F1 | AUC | Acc | Prec | Recall |
| PET | 0.8587 | 0.8569 | 0.8265 | 0.9178 | 0.8465 | 0.9592 | 0.9898 | 0.9615 | 0.9632 | 0.9615 |
| Tabular | 0.7722*** | 0.7776*** | 0.7119*** | 0.8438 | 0.7438*** | 0.8776*** | 0.7582*** | 0.8125*** | 0.9180*** | 0.8125*** |
| $T_1$ and PET | 0.7671 | 0.8243 | 0.8177 | 0.7338 | 0.8177 | 0.9392 | 0.9861 | 0.9375 | 0.9422 | 0.9375 |
| $T_1$ and Tabular | 0.7761*** | 0.8023*** | 0.8021** | 0.7582*** | 0.8021** | 0.8699*** | 0.8975*** | 0.8958*** | 0.9072*** | 0.8958*** |
| PET and Tabular | 0.8163 | 0.8368 | 0.8021 | 0.8393 | 0.8021 | 0.9604 | 0.9910 | 0.9583 | 0.9676 | 0.9682 |
| $T_1$ and PET and Tabular | 0.8541 | 0.8996 | 0.8594 | 0.8504 | 0.8594 | 0.9785 | 0.9952 | 0.9792 | 0.9797 | 0.9715 |
| PET* | 0.8310 | 0.8373 | 0.8265 | **0.8782** | 0.8087 | 0.9547 | 0.9505 | 0.9481 | 0.9602 | 0.9583 |
| PET* and Tabular | 0.7897 | 0.7905 | 0.7756 | 0.8083 | 0.7856 | 0.9301 | 0.9512 | 0.9375 | 0.9418 | 0.9375 |
| $T_1$ and PET* and Tabular | **0.8334** | **0.8377** | **0.8281** | 0.8399 | **0.8281** | **0.9585** | **0.9812** | **0.9503** | **0.9612** | **0.9644** |

Table 3: Comparison of pMCI vs sMCI classification results by different modality data.

To further evaluate the synthesized PET images, we employ a downstream task of classification of progressive Mild Cognitive Impairment (pMCI) and stable Mild Cognitive Impairment (sMCI). Mild Cognitive Impairment (MCI) is a stage before dementia. pMCI is likely to progress to AD while sMCI remains relatively stable over time. Distinguishing between pMCI and sMCI is essential for early intervention and treatment planning. Based on the encoder of the $T_1$ MRI encoder, we train a PET classifier on the real PET images and synthesized PET are only for test. We set ten different random seeds and conducted ten rounds of model training and validation. The mean results are reported in the table, with the best values highlighted in bold, and the $t$-values from paired $t$-tests with other methods are also provided. ***: p < 0.001, **: p < 0.01, and *: p < 0.05. As Table 2 shows, for both the AV45 and AV1451, our method scores the highest in most metrics. Real PET images provide the benchmark and our proposed method closely approximates these results.

Table 3 shows the classification results using different modalities data. PET* denoted the synthesized PET images while PET denoted the real PET images. We set ten different random seeds and conducted ten rounds of model training and validation. We also conducted paired $t$-tests between the multimodal classification results using synthetic PET and those without synthetic PET (using only tabular data, T1 MRI, or their combination). The results demonstrate that the improvement in classification performance with synthetic PET is statistically significant. ***: p < 0.001, **: p < 0.01, and *: p < 0.05. The results indicate the clinical significance of our method.

## 3.4. Interpretability

Generating counterfactual PET images by intervening on variables explains their roles in the model. For example, reducing APOE4 count lowers SUVR in generated PET images ($\Delta n_{APOE4} = -1$), shown in Fig. 4, indicating less amyloid or tau deposition, consistent with clinical findings (Lim et al., 2017). This enhances Causal PETS' interpretability.

These observations are further supported by the regional analysis presented in Table 4. We divided the brain into six regions of interest (ROIs): Frontal Cortex (FC), Temporal Cortex (TC), Parietal Cortex (PC), Occipital Cortex (OC), Cingulate and Insula (CI), and Operculum and Orbital Areas (OO). For each ROI, we performed a paired t-test to compare SUVR values before and after the counterfactual intervention on APOE4 count.

As shown in Table 4, different tracers exhibit distinct regional sensitivity to APOE4 alterations. For AV45, which primarily targets amyloid deposition, the most significant reductions in SUVR when decreasing APOE4 count ($\Delta n_{APOE4} = -1$) occur in FC, PC, and CI ($p < 0.001$), followed by TC and OC ($p < 0.01$)**. This suggests that these regions are particularly susceptible to APOE4-related amyloid accumulation. In contrast, AV1451, which binds to tau pathology, shows relatively weaker effects in some regions, such as OC and CI, but still exhibits significant reductions in FC, TC, and PC. Notably, the operculum and orbital areas (OO) show less pronounced differences in both tracers, potentially indicating lower tracer sensitivity in these regions or reduced APOE4-driven pathology.

Further discussion and additional regional analyses can be found in the appendix.

Table 4: ROI Mean Difference and P-Value

| Mean Difference | FC | TC | PC | OC | CI | OO |
|---|---|---|---|---|---|---|
| **AV45 SUVR** | | | | | | |
| $\Delta_{n_{APOE}} = -1$ | -0.0236*** | -0.0201*** | -0.0235*** | -0.0098*** | -0.0280*** | -0.0164*** |
| $\Delta_{n_{APOE}} = 1$ | 0.0081** | 0.0076*** | 0.0063** | 0.0019 | 0.00956** | 0.0038 |
| **AV1451 SUVR** | | | | | | |
| $\Delta_{n_{APOE}} = -1$ | -0.0222** | -0.0183*** | -0.0021** | 0.0138 | -0.0189 | -0.0295 |
| $\Delta_{n_{APOE}} = 1$ | 0.0028* | 0.0160** | 0.0012 | 0.0144** | 0.0011 | 0.0004 |

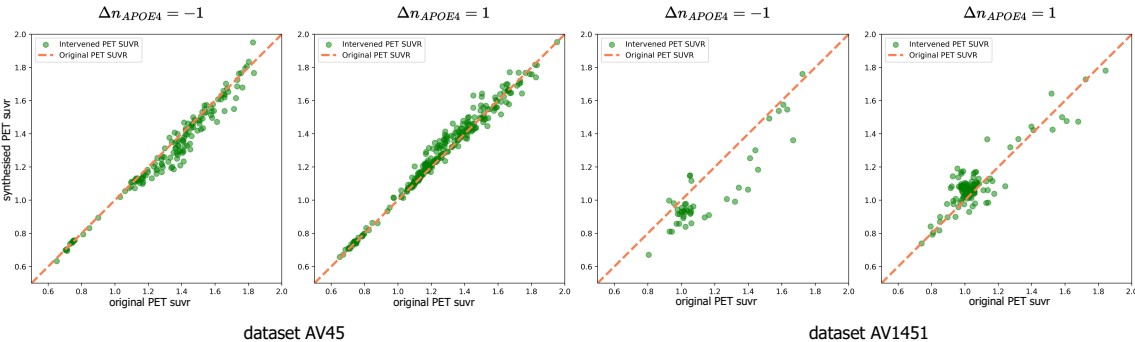

Figure 4: The SURV comparisons of counterfactual PET

## 3.5. Ablation Study

To validate the contribution of each module and loss function component, we conducted ablation experiments, shown in Table 5. Removing $I_{T1}$ reconstruction leads to the highest MAE increase and SSIM drop, while omitting the PET Discriminator also degrades performance, highlighting their roles in image refinement. Additionally, perceptual loss and $\epsilon_{I_{PET}}$ help reduce variance, suggesting that $\epsilon_{I_{PET}}$ improves robustness to image noise.

| Method | dataset AV45 | | | | dataset AV1451 | | | |
|---|---|---|---|---|---|---|---|---|
| | MAE($\times 10^{-1}$) ↓ | SSIM(%) ↑ | PNSR↑ | $\epsilon_{SUVR}$ ↓ | MAE($\times 10^{-1}$) ↓ | SSIM(%) ↑ | PNSR↑ | $\epsilon_{SUVR}$ ↓ |
| w/o $I_{T1}$ reconstruction | $0.247_{\pm 0.109}$ | $97.33_{\pm 1.15}$ | $26.25_{\pm 2.53}$ | $0.130_{\pm 0.120}$ | $0.265_{\pm 0.119}$ | $98.11_{\pm 1.08}$ | $24.02_{\pm 2.24}$ | $0.090_{\pm 0.089}$ |
| w/o PET Discriminator | $0.243_{\pm 0.109}$ | $97.29_{\pm 1.10}$ | $26.25_{\pm 2.49}$ | $0.136_{\pm 0.110}$ | $0.235_{\pm 0.102}$ | $98.01_{\pm 0.93}$ | $26.70_{\pm 1.96}$ | $0.091_{\pm 0.094}$ |
| w/o Perceptual loss | $0.233_{\pm 0.111}$ | $97.31_{\pm 1.10}$ | $26.46_{\pm 2.61}$ | $0.124_{\pm 0.120}$ | $0.211_{\pm 0.092}$ | $97.28_{\pm 0.62}$ | $28.26_{\pm 2.07}$ | $0.083_{\pm 0.104}$ |
| w/o $\epsilon_{I_{PET}}$ | $0.233_{\pm 0.131}$ | $97.30_{\pm 1.31}$ | $26.36_{\pm 2.85}$ | $0.131_{\pm 0.182}$ | $0.212_{\pm 0.122}$ | $97.28_{\pm 1.49}$ | $27.38_{\pm 2.38}$ | $0.089_{\pm 0.133}$ |
| Ours | $\mathbf{0.224_{\pm 0.104}}$ | $\mathbf{97.47_{\pm 1.13}}$ | $\mathbf{26.740_{\pm 2.581}}$ | $\mathbf{0.10_{\pm 0.13}}$ | $\mathbf{0.202_{\pm 0.096}}$ | $\mathbf{98.12_{\pm 0.82}}$ | $\mathbf{29.687_{\pm 1.905}}$ | $\mathbf{0.08_{\pm 0.10}}$ |

Table 5: Ablation Study on AV45 and AV1451 Dataset

## 4. Conclusion

In this work, we propose **Causal PETS**, a novel causality-informed synthesis model for generating PET images from multi-modal data. Our model analyzes the causal relationships between different modalities to generate PET images. Our causality-informed PET

synthesis model represents a significant step forward in the integration of multi-modal data for medical imaging. However, our work still has some limitations, e.g., we do not consider the temporal dimension. By addressing the limitations we can enhance the clinical applicability and impact of this approach.

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

## Appendix A. Dataset

AV45 PET imaging (Johnson et al., 2013), also known as florbetapir (AV-45) PET imaging, is used to visualize amyloid plaques. Correspondingly, CSF $A\beta$ data is chosen as $B_{CSF}$, referring to measurements of amyloid-beta peptides in the CSF. AV45 PET images and CSF $A\beta$ data reflect the amyloid pathology in brain. AV1451 PET imaging (Mishra et al., 2017), also known as flortaucipir (AV-1451)PET imaging, is the imaging of tau protein tangles. Meanwhile, CSF tau and p-tau data measure the total tau and phosphorylated tau proteins in the CSF, respectively. They both reflect the tau pathology.

The dataset details for training the synthesis model is as follows. We divided the dataset into training, validation, and test sets in a 4:1:1 ratio, ensuring that all scans from the same individual appear in the same set, thereby preventing data leakage.

| AV45 dataset | | | |
|---|---|---|---|
| Category | CN | MCI | AD |
| # of subjects | 519 | 476 | 197 |
| # of sessions | 776 | 954 | 213 |
| Age | 74.84±7.29 | 73.29±7.44 | 73.60±7.46 |
| **AV1451 dataset** | | | |
| Category | CN | MCI | AD |
| # of subjects | 187 | 130 | 61 |
| # of sessions | 314 | 269 | 82 |
| Age | 72.96±7.32 | 72.80±7.08 | 72.73±8.02 |

Table 6: The Basic Information of ADNI Dataset

For imaging processing, all $T_1$ MRI are skull-stripped using ROBEX (Iglesias et al., 2011), aligned to the MNI152 space, resampled to 1.5mm isotropic using ANTs (Avants et al., 2009), cropped to dimensions of 96×128×96, and normalized in voxel values to range [0, 1]. The PET images are skull-stripped using ROBEX (Iglesias et al., 2011), registered to the paired $T_1$ MRI, normalized to range [0, 1].

To pair $I_{PET}$ with $I_{T1}$ and $B_{CSF}$, we define a successful pairing as having a measurement interval of less than 6 months. let $t_{PET}$, $t_{T1}$, and $t_{CSF}$ represent the time points at which the data were taken. A successful pair is defined as:

$$\max\left(|t_{PET} - t_{T1}|, |t_{PET} - t_{CSF}|, |t_{T1} - t_{CSF}|\right) < 6 \text{ months.}$$

This condition ensures that the measurements $I_{PET}$, $I_{T1}$, and $B_{CSF}$ are temporally aligned.

## Appendix B. Training Process

We adopt an adversarial training approach, where the generator and discriminator are alternately trained. The ADAM optimizer is used with a learning rate (LR) of 0.0001 for the generator, fusion Network, auto-encoder and a LR of 0.0005 for the discriminator. The training is conducted over 1000 epochs, taking approximately 1.5 days. The batch size is set to 2, and we use 6 NVIDIA TITAN RTX GPUs for parallel training. All code is

implemented based on PyTorch. The auto-encoder, generator, and discriminator are built using the basic architectures from MONAI. The $\lambda_{Perceptual}$ is set to 0.02 and $\lambda_{adv}$ is set to 0.005 in the overall loss function.

## Appendix C. SUVR formula

The SUVR value is calculated using the following formula:

$$\text{SUVR} = \frac{\frac{1}{|V_{\text{ROI}}|} \sum_{v \in V_{\text{ROI}}} v}{\frac{1}{|V_{\text{ref}}|} \sum_{v \in V_{\text{ref}}} v}, \tag{1}$$

where $V_{\text{ROI}}$ is the set of voxel values in ROI, and $V_{\text{ref}}$ is the set of voxel values in the reference region. $|V_{\text{ROI}}|$ and $|V_{\text{ref}}|$ denote the number of voxels in each respective region. As recommended in clinical research (Schindler et al., 2021; Jack Jr et al., 2018), the cerebral cortex region is set as the ROI.

## Appendix D. Extended Experiments of AD classification

| Method | dataset AV45 | | | | | dataset AV1451 | | | | |
|---|---|---|---|---|---|---|---|---|---|---|
| | F1 | AUC | Acc | Prec | Recall | F1 | AUC | Acc | Prec | Recall |
| Uni-Modal | | | | | | | | | | |
| CycleGAN (Zhu et al., 2017) | 0.5360*** | 0.5852*** | 0.5080*** | 0.7260*** | 0.5080*** | 0.6877*** | 0.5061*** | 0.6742*** | 0.7019*** | 0.6742*** |
| Pix2Pix (Isola et al., 2017b) | 0.7627*** | 0.7231*** | 0.7968*** | 0.7768*** | 0.7968*** | 0.7810*** | 0.5455*** | 0.8427*** | 0.7278*** | 0.8427*** |
| Unet w/o condition | 0.7938*** | 0.8552*** | 0.8075*** | 0.7920*** | 0.8075*** | 0.7983*** | 0.8102*** | 0.7774*** | 0.8440*** | 0.7774*** |
| Multi-Modal(Attention-based Fusion) | | | | | | | | | | |
| Unet w/ condition | 0.8140** | 0.8522* | 0.8021** | **0.8500** | 0.8021*** | 0.8453*** | 0.8596*** | 0.8468*** | 0.8440*** | 0.8468*** |
| BMGAN (Hu et al., 2021) | 0.7277*** | 0.7517*** | 0.7914*** | 0.8049*** | 0.7914*** | 0.8410*** | 0.8261*** | 0.8597*** | 0.8433*** | 0.8597*** |
| BPGAN (Zhang et al., 2022b) | 0.7594*** | 0.7087*** | 0.7754*** | 0.7538*** | 0.7754*** | 0.7983*** | 0.8102*** | 0.7774*** | 0.8440*** | 0.7774*** |
| ControlNet (Zhang et al., 2023) | 0.7588*** | 0.7431*** | 0.7692*** | 0.7590*** | 0.7692*** | 0.8120*** | 0.6957*** | 0.7600*** | **0.9116** | 0.7600*** |
| CLDM (Ou et al., 2024a) | 0.5980*** | 0.5903*** | 0.6154*** | 0.5872*** | 0.6154*** | 0.6714*** | 0.7066*** | 0.6274*** | 0.8583*** | 0.6274*** |
| PASTA (Li et al., 2024) | 0.6272*** | 0.6389*** | 0.6538*** | 0.6176*** | 0.6538*** | 0.7928*** | 0.7279*** | 0.7692*** | 0.8234*** | 0.7692*** |
| Multi-Modal(Causality-based Fusion) | | | | | | | | | | |
| **CausalPETS (ours)** | **0.8354** | **0.8703** | **0.8289** | 0.8492 | **0.8289** | **0.8720** | **0.8824** | **0.8926** | 0.9050* | **0.8926** |
| Real Images | 0.8472 | 0.8996 | 0.8396 | 0.8677 | 0.8396 | 0.9072 | 0.9295 | 0.9032 | 0.9153 | 0.9032 |

Table 7: Comparison of CN vs AD classification results using synthesised PET images.

**AD Classification Results.** In addition to the pMCI vs. sMCI classification experiments shown in the main paper, we further evaluate the quality of synthesized PET images on the downstream task of AD vs. CN classification. As shown in Table 7, this task involves a larger population and presents a more balanced and robust evaluation of model generalization. We compare our method with several representative baselines under three settings: Uni-Modal (using only synthetic PET), Multi-Modal with attention-based fusion, and Multi-Modal with causality-based fusion. Our method consistently outperforms all baselines across both datasets (AV45 and AV1451) and all evaluation metrics (F1, AUC, Accuracy, Precision, and Recall). The strong performance in this more general classification setting confirms the reliability and superiority of our synthesized images for supporting clinical-level decision-making tasks.

## Appendix E. Interpretability

In this section we generate PET images with intervening on one of the variables in Fig. 1, and the role of each variable within the model can be explained. This counterfactual manipulation allows us to explore the role of variables in our causal PETS model, thereby enhancing the interpretability of the model.

Here we presents the visualisation results of synthesised PET with the intervention on $B_{CSF}$ ($A\beta_{42}$ for AV45 dataset and $p\tau_{181}$ for AV1451 daraset).

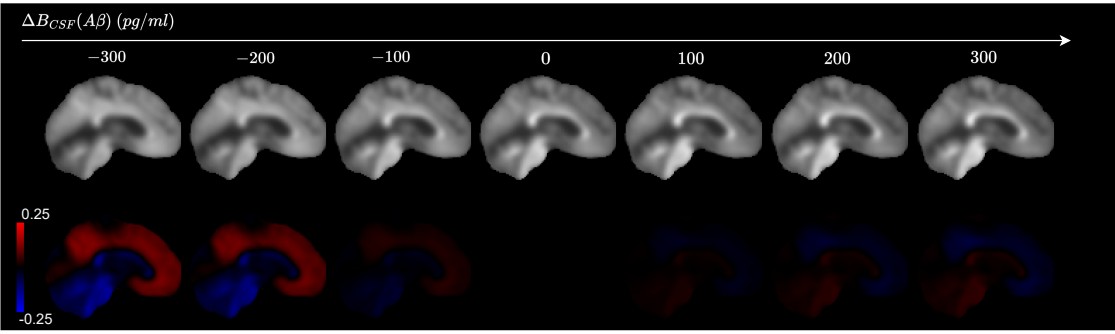

Figure 5: The PET images generated by intervening on the $B_{CSF}(A\beta_{42})$, experiments on AV45 dataset. The first row shows the generated PET image and the second row shows the difference map.

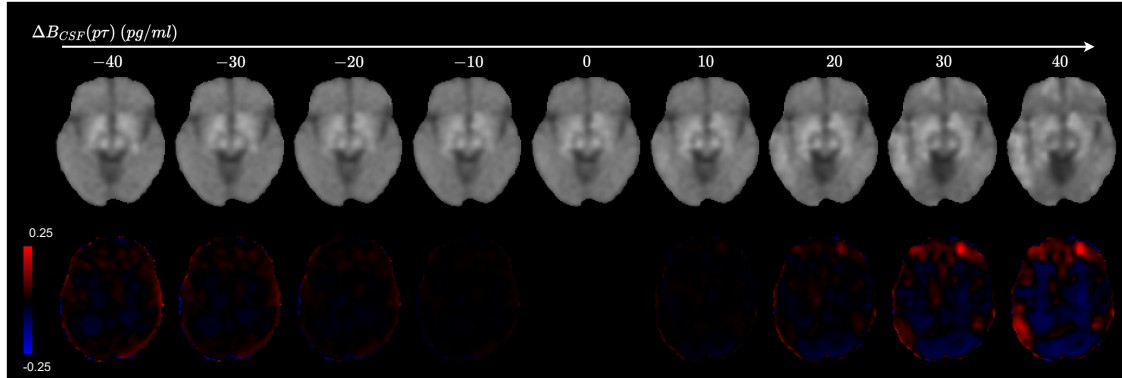

Figure 6: The PET images generated by intervening on the $B_{CSF}(p\tau)$, experiments on AV1451 dataset. The first row shows the generated PET image and the second row shows the difference map.

In dataset AV45, $B_{CSF}$ measures the soluble $A\beta_{42}$ concentration in CSF, while PET detects the deposited amyloid in the brain. Lower levels of $A\beta_{42}$ in the CSF are often associated with higher levels of amyloid deposition in the brain because $A\beta_{42}$, which is a form of amyloid-beta, tends to accumulate in amyloid plaques in the brain, leading to reduced levels in the CSF. Thus, a lower CSF concentration indicates more amyloid deposition in the brain, resulting in a higher-signal in PET image, and vice versa. As Fig. 5 shows, our

visualization results reflect the specific locations (mostly cerebral cortex) and patterns of amyloid deposition in the brain as CSF concentration decreases.

In dataset AV1451, $B_{CSF}$ measures $p\tau_{181}$ protein and binds to neurofibrillary tangles, which are aggregates of $p\tau$ associated with AD. The $p\tau_{181}$ protein is primarily generated in the brain and enters the CSF through the blood-brain barrier. Thus, a lower CSF concentration indicates less $p\tau_{181}$ in the brain, resulting in a lower-signal in PET image, and vice versa. As Fig. 6 shows, our visualization results reflect the specific locations and patterns of $p\tau_{181}$ in the brain as CSF concentration increases.

