# OpenReview forum: "Causal PETS: Causality-Informed PET Synthesis from Multi-modal Data"
_MIDL.io/2025/Conference — MIDL 2025 Poster_

### Official Review · Reviewer_xGdK · 2025-02-19

**Confidence:** 5
**Preliminary Rating:** 2
**Final Rating:** 4

**Summary:**

This paper proposes an MRI-to-PET synthesis method based on causal graph information + GAN from other modalities (CSF, demographic information, APOE4 genetic data).

**Strengths:**

1. The paper provides a detailed comparison between the generative model performance and the downstream classification tasks.
2. The paper offers a clear and concise description of the proposed method, the text is well-organized, and the figures in the paper are presented in a highly effective manner.

**Weaknesses:**

1. the additional modality information introduced (such as the APOE4 gene and CSF biomarkers) is not as easily accessible as PET, and its actual clinical value is unclear.
2.  I do not see any significant advancements compared to the authors' previous work [1] in 2023.
3.  In terms of experimental evaluation, the choice of MAE/PSNR performance metrics is not appropriate for assessing the SUV ratio characteristics, and the APOE-related interpretability results fail to provide meaningful conclusions.

[1] Li Y, Shi J, Zhou S K. Causal image synthesis of brain mr in 3d[J]. arXiv preprint arXiv:2303.14349, 2023.

**Detailed Comments:**

(1) Inappropriate Input for Clinical Scenarios.  MRI-to-PET synthesis is generally intended to address the challenge of inaccessible or missing modalities. However, the APOE4 and CSF biomarkers are also not easily accessible and, in fact, are harder to obtain than PET. If clinicians have access to APOE4, CSF biomarkers, and T1WI MRI, they can already make diagnostic decisions with high confidence, making the need for this synthesis method unclear.

(2) Limited Novelty:

(i) The proposed method lacks novelty, as the core causal image synthesis approach was already introduced in the authors' previous work [1]. It appears that this paper is simply applying the same method to a different task.

(ii) From a generative perspective, GAN-based methods have been in use for over a decade (since 2014), and many diffusion-based variations now provide high-fidelity, diverse performance. Recently, visual autoregressive models have demonstrated impressive generative capabilities as well.

(iii) Why not employ an attention-based module to construct the causal graph without introducing excessive inductive bias? The edge information appears insufficient; for instance, the distribution of biomarkers (B) should influence brain structure (S).

(3) Unfair and Insufficient Results:

(i) MAE and PSNR are not appropriate metrics for multi-tracer PET evaluation, as PET SUVR is a ratio dependent on the radiotracer type, not a fixed voxel value like traditional natural images (0-255). For example, AV1451 PET showing better MAE/PSNR performance than AV45 PET does not indicate that AV1451 produces better synthetic performance.

(ii) The comparison results in Table 1 are inconsistent with the proposed method. Were the same network parameters used for the evaluation? Does there exist an overlap between the training and testing subjects?

(iii) The improvement in classification results is limited. The authors need to conduct more significant statistical evaluations to demonstrate significant performance improvements in the proposed method.

(iv) The APOE4-related analysis in Fig. 4 is insufficient and lacks meaningful insights. In reality, different radiotracers will select tracer-specific meta-ROIs (regions of interest for clinical evaluation), and these meta-ROIs' SUVR values will differ significantly between APOE4 carriers and non-carriers. It should be noted that overall per-voxel SUVR evaluation is not meaningful in the same way as per-voxel SUVR MAE/PSNR.

[1] Li Y, Shi J, Zhou S K. Causal image synthesis of brain mr in 3d[J]. arXiv preprint arXiv:2303.14349, 2023.

**Justification Of The Final Rating:**

The authors included additional downstream evaluations for NC/AD, provided substantial evidence, and validated APOE4 stratification. The current rebuttal conclusion is adequate and demonstrates significant clinical value. Therefore, I recommend a weak acceptance of this paper.

**Justification Of The Preliminary Rating:**

The clinical value or motivation is unclear, the technical details are out-of-data that GAN was published in 2014, and the experimental results need more extensive modification or evaluation to show the effectiveness of the proposed method.

**Questions To Address In The Rebuttal:**

1. the clinical value or motivation needs to be clarified again.
2. the statistical evaluation results for all the classification results show a significant improvement in the proposed method.
3. the core technical contributions or differences with the previous work in 2023 are needed.
4. More analysis of APOE4-related results is needed.

---

> ### Author Response · Authors · 2025-03-07
> **We clarified the clinical scenarios and the novelty limitations, and added meta-ROI-based analysis.**
>
> 1.  Inappropriate Input for Clinical Scenarios
>
> (1) **PET provides unique local pathological information that APOE4/CSF cannot replace**. PET directly provides the spatial distribution of beta-amyloid (Aβ) and tau pathology in the brain, whereas APOE4 is merely a genetic risk factor, and CSF biomarkers reflect global pathological states rather than localized brain pathology. Even when clinicians have APOE4 and CSF data, they still rely on PET imaging to confirm the spatial patterns of pathological burden. For example: APOE4 carriers do not always develop AD, and their CSF Aβ and tau level vary significantly between individuals. **PET imaging provides more direct and interpretable evidence of disease progression.**
>
> (2) In fact, our initial plan was to incorporate biomarker concentrations from peripheral blood tests as variables, as they are more accessible, non-invasive, and cost-effective, making them a more suitable choice for our setting compared to CSF. However, most publicly available datasets currently lack this variable. **Nevertheless, our work can seamlessly extend to scenarios involving peripheral blood test samples in the future.**
>
> 2. Limited Novelty
>
> (1) **The previous work [1] was not published in any peer-reviewed conference or journal and primarily served as an initial exploration.** This paper presents a substantial expansion and refinement of that preliminary idea, incorporating new methodological advancements, additional datasets, stronger causal modeling constraints, and more comprehensive evaluations.
>
> While this work builds upon [1], it introduces several critical innovations that go beyond a mere application to a different task:
>
> (1 **New causal graph design**: The causal structure is significantly revised to accommodate multi-modal dependencies and disease-specific pathophysiology, which were not considered in [1].
> (2 **Rigorous evaluation on clinically meaningful tasks**: Unlike [1], which focused solely on counterfactual image synthesis quality, this study provides clinical-task-driven validation, including downstream classification, and biomarker association analysis, ensuring practical utility. **
>
> We sincerely appreciate the insightful suggestions about more advanced generative models and attention mechanism and will incorporate them into our future research. **
>
> [1] Li Y, Shi J, Zhou S K. Causal image synthesis of brain mr in 3d[J]. arXiv preprint arXiv:2303.14349, 2023.
>
> 3. Unfair and Insufficient Results
>
> (1) Regarding the appropriateness of MAE and PSNR for multi-tracer PET evaluation, **MAE and PSNR remain widely used in medical image synthesis to assess pixel-level fidelity**. It’s true that across different tracers, better MAE/PSNR performance does not indicate better synthetic performance. **But, for a single scan (same subject and tracer), better MAE/PSNR performance can the superiority of the model in a specific aspect and we perform a pair t-test to demonstrate this. **
> **Besieds, we also report $Epsilon_{SUVR}$, which directly evaluates SUVR error, making it a more relevant metric for PET synthesis analyze the downstream task performance (e.g., sMCI vs. pMCI classification), which indirectly reflects the clinical relevance of the synthetic images.**
>
> (2) We confirm that all comparison methods used the same network architecture and hyperparameters, ensuring a fair evaluation and there is no data leakage between training and testing sets. We strictly separate subjects between train and test sets. The code has also been available to reproduce the results.
>
> (3)We acknowledge that the observed classification improvements are modest. **To further support our claims, we have performed statistical significance testing (paired t-tests) to quantify improvements, please refer to Table1, 2, 3.**
>
> (4)We appreciate the reviewer’s suggestion regarding meta-ROI-based analysis. **We have done such experiments in the revised manuscript, please refer to Table 5.**

---

> ### Comment · Reviewer_xGdK · 2025-03-07
>
> I truly appreciate the authors' efforts and the detailed responses provided. I genuinely wanted to give a higher score to support the paper’s acceptance, as it presents an interesting exploration. However, after careful consideration and reasoning, I believe a “weak reject” is the most appropriate recommendation I can justify.
>
> Below are my specific responses:
>
> **Reply to A1:** I fully agree that “real PET” is crucial for providing direct visual evidence of disease progression, as the authors point out. However, what I want to emphasize is that the clinical value or confidence clinicians might place in “generated PET” (which are faked) is not significantly greater than the clinical confidence placed in “real CSF + real APOE4 data.” Furthermore, the costs associated with CSF and genetic data are not less than those for real PET.
>
> **Reply to A2:** In terms of the conference review standards, I have three main concerns: task type, technical novelty, and experimental evaluation.
> i) The MRI-to-PET task itself is not novel.  ii) Even after the authors clarified the comparison with previous work, I still feel that the improvements are limited. Whether or not the previous work was an initial exploration, it was proposed in 2023. Moreover, the technical combination of 'causal graphs + GANs' lacks significant novelty and seems more like an incremental advancement. iii) The evaluation is thorough but not particularly convincing in some aspects.
>
> **Reply to A3:** While the evaluation is thorough, I must admit that after considerable reflection, I remain unconvinced about the authenticity of some of the experimental results in this paper. As someone working in the cross-modal synthesis field and having conducted AD-related diagnostic tasks, I find the results—particularly the SMCI and PMCI results in Table 3—somewhat implausible from both a neurological and computational perspective, especially considering the performance changes that come with adding more medical modalities.

---

> > ### Author Response · Authors · 2025-03-08
> >
> > Reply to A1: Clinical Value of Generated PET
> >
> > This work was initiated by a hospital and has been conducted with continuous involvement from clinicians. The purpose of this project has never been to replace real PET scans with generated images but rather to provide valuable insights from a causal perspective to assist doctors. In our clinical application, doctors do not simply state, "Your brain will look like this in the future." Instead, they use our approach as a reference to improve their diagnoses, such as determining optimal medication and dosage for different patients at different stages.
> >
> > **Furthermore, our technical approach has the potential for further development**. We introduce a causality-driven generative model rather than a standard image-to-image translation model, which can be applied to other medical scenarios.
> >
> > Reply to A2: Task Novelty, Technical Contribution, and Evaluation
> >
> > (i) Task Novelty: We acknowledge that MRI-to-PET synthesis has been explored before. However, our work is distinct in that we introduce a causality-driven generative model rather than a standard image-to-image translation model. **This provides a new perspective on medical image synthesis by explicitly modeling causal relationships in disease progression.**
> >
> > (ii) Technical Contribution: We integrate causal reasoning with diffusion models in a way that enables more interpretable and controllable PET synthesis. Prior work in this domain has largely focused on black-box generative models without explicitly modeling causality, making our approach fundamentally different.
> >
> > Reply to A3: Plausibility of Experimental Results
> >
> > We were also surprised by the strong classification performance on AV1451, whereas the results for AV45 were more in line with expectations. However, the experimental results remain as they are. **In the ADNI dataset, the number of AV1451 samples meeting our inclusion criteria is relatively small, with a total of 130 MCI subjects, including 18 sMCI and 5 pMCI cases in the test set. The limited sample size makes it possible for the test performance to appear exceptionally high.** In contrast, the AV45 dataset contains more than three times as many samples as AV1451. Moreover, although the AV1451 dataset is relatively small, our model still demonstrates significantly better performance compared to other models. We will further validate our model’s performance on larger datasets in the future.

---

> > > ### Comment · Reviewer_xGdK · 2025-03-08
> > > **Meaningless downstream task evaluation**
> > >
> > > Based on the small downstream dataset, where the baseline performance is 78% (18/23), I firmly believe that the downstream analysis is not meaningful for evaluating such multi-modal classification task.

---

> > > > ### Author Response · Authors · 2025-03-09
> > > >
> > > > (1) While the AV1451 dataset is relatively small, it still includes 130 MCI subjects and over 500 scans in total. Besides, We have conducted multiple experiments to confirm the stability of our model.
> > > >
> > > > (2) **Since ADNI has released this PET modality, we believe it is worth applying our method to it, despite its size. Public PET datasets are inherently scarce and often small in scale.** ADNI also provides PET data for other tracers, some of which have fewer than 50 subjects. Given this, we have already chosen one of the larger available datasets.
> > > >
> > > > (3) **Additionally, we have a larger AV45 dataset, and the smaller AV1451 dataset serves as a valuable complementary source of information. The primary goal of using AV1451 is not solely classification performance but also to demonstrate the applicability of our approach across different PET tracers.** Furthermore, in other tasks such as image synthesis quality and interpretability experiments, the AV1451 dataset has not posed any issues.

---

> > > > > ### Comment · Reviewer_xGdK · 2025-03-10
> > > > >
> > > > > Since deep learning is a data-driven technique, the training dataset size must be appropriately aligned with the complexity of the research problem. From my perspective, the limited size of the downstream dataset, particularly the small testing set, is insufficient to provide any meaningful performance insights, even when multiple runs with different random seeds are conducted. I strongly recommend that the authors re-evaluate the downstream task using a significantly larger NC/AD classification dataset. The current downstream evaluation does not effectively demonstrate the diagnostic effectiveness of the proposed cross-modal synthesis method. Consequently, I will lower my score and recommend a "reject."

---

> > > > > > ### Author Response · Authors · 2025-03-10
> > > > > >
> > > > > > **We genuinely want to extract useful feedback to improve our work.** However, after thoroughly reviewing your comments, we find no meaningful or constructive advice that could help us enhance our paper.
> > > > > >
> > > > > > First, regarding the dataset concerns—this dataset is publicly available. If you believe the dataset is inadequate, that is a limitation of the publicly available medical imaging data, not a flaw in our methodology. Nevertheless, we acknowledge your concern and are happy to explicitly report in our paper that an MIDL reviewer finds this dataset meaningless for DL research.
> > > > > >
> > > > > > If you believe a more suitable dataset exists, we would greatly appreciate it if you could suggest one. However, if we were to exclude this dataset entirely, the significance of the study would be even further diminished, given the limited availability of multi-modal PET datasets.
> > > > > >
> > > > > > Finally, we would like to emphasize that MIDL is a venue focused on technical contributions, not purely clinical research. If we were to frame our work solely from a clinical perspective, it would be difficult to fit within the scope of MIDL.

---

> > > > > > > ### Comment · Reviewer_xGdK · 2025-03-10
> > > > > > > **Final Response**
> > > > > > >
> > > > > > > I feel that the author has misunderstood some of my expressions, which led to the perception that I lack respect for the dataset:
> > > > > > >
> > > > > > > **Firstly, without the ADNI database, there would be no development for us or for the AD field over the years.** I have always had great respect and gratitude for the builders of the ADNI database. At the same time, the ADNI database requires all users of the corresponding dataset to explicitly list ADNI in the author list. However, it seems the author has not adhered to the ADNI guidelines [1].
> > > > > > >
> > > > > > > **Secondly, I believe the small sample size of the sMCI/pMCI in the downstream task evaluation (with only 23 samples in the test set and only 5 negative samples) fails to reflect the diagnostic effectiveness of the proposed method.** The reasons are as follows:
> > > > > > >
> > > > > > > a. The sMCI and pMCI sample sizes in the dataset are sufficient (there are more than 1200 MCI subjects, currently including ADNI4 [2]). The reason for the smaller dataset in this paper is **due to the strong requirement for clinical data, including CSF and APOE4, among other types of inputs**. The group with complete input data is smaller.
> > > > > > >
> > > > > > > b.In contrast, similar works that use MCI for evaluation have already included 1080 [3] and 500 [4] subjects. I must emphasize that **the small evaluation dataset is not a problem with the dataset itself but a result of the strict input conditions for the model.**
> > > > > > > This further highlights the limited clinical value of the method.
> > > > > > >
> > > > > > > c. The effective solution I propose is to **use a larger population of NC/AD patients**, which has been widely applied in downstream evaluation in cross-modal synthesis [4-6]. With a larger patient sample, the downstream tasks can be more robust, rather than being **highly sensitive to a few misclassified samples** that cause significant performance changes.
> > > > > > >
> > > > > > > As previously stated, my evaluation was based on clinical value, technical innovation, and experiments. The score correction was made after a thorough, objective reflection on the robustness of the experimental results, which led to my modification of the score.
> > > > > > >
> > > > > > > The author may consider my suggestions as not constructive, and **I am willing to apologize for any inappropriate expressions. However, I can honestly say that I approached my role as a reviewer with an objective and calm attitude when providing my feedback and reasonably modifying the score.** This will be my final response. A polite rebuttal and the ability to view our own work from a third-person perspective is something every researcher should strive for. I sincerely wish the author success in their future research.
> > > > > > >
> > > > > > > [1] https://adni.loni.usc.edu/wp-content/uploads/how_to_apply/ADNI_DSP_Policy.pdf
> > > > > > >
> > > > > > > [2] https://adni.loni.usc.edu/about/
> > > > > > >
> > > > > > > [3] Ou Z, Jiang C, Liu Y, et al. A Graph-Embedded Latent Space Learning and Clustering Framework for Incomplete Multimodal Multiclass Alzheimer’s Disease Diagnosis[C]//International Conference on Medical Image Computing and Computer-Assisted Intervention. Cham: Springer Nature Switzerland, 2024: 45-55.
> > > > > > >
> > > > > > > [4] Chen Y, Pan Y, Xia Y, et al. Disentangle first, then distill: a unified framework for missing modality imputation and Alzheimer’s disease diagnosis[J]. IEEE Transactions on Medical Imaging, 2023, 42(12): 3566-3578.
> > > > > > >
> > > > > > > [5] Li Y, Yakushev I, Hedderich D M, et al. PASTA: P athology-A ware MRI to PET Cro S s-modal T r A nslation with Diffusion Models[C]//International Conference on Medical Image Computing and Computer-Assisted Intervention. Cham: Springer Nature Switzerland, 2024: 529-540.
> > > > > > >
> > > > > > > [6] Pan, Yongsheng, et al. "Disease-image-specific learning for diagnosis-oriented neuroimage synthesis with incomplete multi-modality data." IEEE transactions on pattern analysis and machine intelligence 44.10 (2021): 6839-6853.

---

> > > > > > > > ### Author Response · Authors · 2025-03-14
> > > > > > > > **AD vs CN results**
> > > > > > > >
> > > > > > > > Thanks for the suggestions and apologize for the agressive response. Nevertheless, we add the AD vs CN experiments, and we believe causal medical image synthesis is a promising direction. We can extend our work with missing modality methods in the future to mitigate the problem of limited data.
> > > > > > > >
> > > > > > > > ### Comparison of CN vs AD classification results using synthesised PET images
> > > > > > > >
> > > > > > > > | Method | F1 | AUC | Acc | Prec | Recall | F1 | AUC | Acc | Prec | Recall |
> > > > > > > > |--------|----|-----|-----|------|--------|----|-----|-----|------|--------|
> > > > > > > > |  | **Dataset AV45** | | | | | **Dataset AV1451** | | | | |
> > > > > > > > | **Uni-Modal** | | | | | | | | | | |
> > > > > > > > | CycleGAN [1] | 0.5360$^{***}$ | 0.5852$^{***}$ | 0.5080$^{***}$ | 0.7260$^{***}$ | 0.5080$^{***}$ | 0.6877$^{***}$ | 0.5061$^{***}$ | 0.6742$^{***}$ | 0.7019$^{***}$ | 0.6742$^{***}$ |
> > > > > > > > | Pix2Pix [2] | 0.7627$^{***}$ | 0.7231$^{***}$ | 0.7968$^{***}$ | 0.7768$^{***}$ | 0.7968$^{***}$ | 0.7810$^{***}$ | 0.5455$^{***}$ | 0.8427$^{***}$ | 0.7278$^{***}$ | 0.8427$^{***}$ |
> > > > > > > > | Unet w/o condition | 0.7938$^{***}$ | 0.8552$^{**}$ | 0.8075$^{***}$ | 0.7920$^{***}$ | 0.8075$^{***}$ | 0.7983$^{***}$ | 0.8102$^{***}$ | 0.7774$^{***}$ | 0.8440$^{***}$ | 0.7774$^{***}$ |
> > > > > > > > | **Multi-Modal (Attention-based Fusion)** | | | | | | | | | | |
> > > > > > > > | Unet w/ condition | 0.8140$^{**}$ | 0.8522$^{*}$ | 0.8021$^{**}$ | **0.8500** | 0.8021$^{***}$ | 0.8453$^{***}$ | 0.8596$^{***}$ | 0.8468$^{***}$ | 0.8440$^{***}$ | 0.8468$^{***}$ |
> > > > > > > > | BMGAN [3] | 0.7277$^{***}$ | 0.7517$^{***}$ | 0.7914$^{***}$ | 0.8049$^{***}$ | 0.7914$^{***}$ | 0.8410$^{***}$ | 0.8261$^{***}$ | 0.8597$^{***}$ | 0.8433$^{***}$ | 0.8597$^{***}$ |
> > > > > > > > | BPGAN [4] | 0.7594$^{***}$ | 0.7087$^{***}$ | 0.7754$^{***}$ | 0.7538$^{***}$ | 0.7754$^{***}$ | 0.7983$^{***}$ | 0.8102$^{***}$ | 0.7774$^{***}$ | 0.8440$^{***}$ | 0.7774$^{***}$ |
> > > > > > > > | ControlNet [5] | 0.7588$^{***}$ | 0.7431$^{***}$ | 0.7692$^{***}$ | 0.7590$^{***}$ | 0.7692$^{***}$ | 0.8120$^{***}$ | 0.6957$^{***}$ | 0.7600$^{***}$ | **0.9116** | 0.7600$^{***}$ |
> > > > > > > > | CLDM [6] | 0.5980$^{***}$ | 0.5903$^{***}$ | 0.6154$^{***}$ | 0.5872$^{***}$ | 0.6154$^{***}$ | 0.6714$^{***}$ | 0.7066$^{***}$ | 0.6274$^{***}$ | 0.8583$^{***}$ | 0.6274$^{***}$ |
> > > > > > > > | PASTA [7] | 0.6272$^{***}$ | 0.6389$^{***}$ | 0.6538$^{***}$ | 0.6176$^{***}$ | 0.6538$^{***}$ | 0.7928$^{***}$ | 0.7279$^{***}$ | 0.7692$^{***}$ | 0.8234$^{***}$ | 0.7692$^{***}$ |
> > > > > > > > | **Multi-Modal (Causality-based Fusion)** | | | | | | | | | | |
> > > > > > > > | **CauFusion (ours)** | **0.8354** | **0.8703** | **0.8289** | 0.8492 | **0.8289** | **0.8720** | **0.8824** | **0.8926** | 0.9050$^{*}$ | **0.8926** |
> > > > > > > > | **Real Images** | 0.8472 | 0.8996 | 0.8396 | 0.8677 | 0.8396 | 0.9072 | 0.9295 | 0.9032 | 0.9153 | 0.9032 |
> > > > > > > >
> > > > > > > > #### References:
> > > > > > > > [1] CycleGAN
> > > > > > > > [2] Pix2Pix
> > > > > > > > [3] BMGAN
> > > > > > > > [4] BPGAN
> > > > > > > > [5] ControlNet
> > > > > > > > [6] CLDM
> > > > > > > > [7] PASTA
> > > > > > > >
> > > > > > > > *Table: Comparison of CN vs AD classification results using synthesised PET images.*

---

> > > > > > > > > ### Comment · Reviewer_xGdK · 2025-03-15
> > > > > > > > > **Weak  Acceptance of the Current Rebuttal**
> > > > > > > > >
> > > > > > > > > The authors included additional downstream evaluations for NC/AD, provided substantial evidence, and validated APOE4 stratification. The current rebuttal conclusion is adequate and demonstrates significant clinical value. Therefore, I recommend a weak acceptance of this paper.
> > > > > > > > >
> > > > > > > > > Additional suggestions: Replace the CN/AD results in the main manuscript with the previous sMCI/pMCI results and move the sMCI/pMCI results to the supplementary materials. Additionally, the current CN/AD results show that tau PET (AV1451) has greater diagnostic and staging capabilities regarding AD progression compared to amyloid PET (AV45), which aligns with real clinical scenarios.

---

### Official Review · Reviewer_gL8J · 2025-02-20

**Confidence:** 4
**Preliminary Rating:** 4
**Recommendation:** Poster
**Final Rating:** 4

**Summary:**

In this work, they present a novel causality-informed multi-modal synthesis model for PET image generation. Here, they explicitly model causal relationships between available multimodal data (MRI, demographics, cerebrospinal fluid (CSF) biomarkers) to generate PET images. They show superior performance in image generation and classification of early diagnosis and monitoring of Alzheimer's disease (AD).

**Strengths:**

- Solid and novel idea on how to use a causal graph of PET synthesis for a generative neural network by approximating the posterior of the brain structure (S) by a VAE and the biomarker distribution (B) by a fusion MLP.
- The work is well motivated by the significant cost of radiotracers and the radiation exposure of PET imaging.
- The additional interpretability provided by the proposed architecture is promising. The generation of counterfactual PET images by intervening on causal variables has the potential to advance scientific discovery.

**Weaknesses:**

- Table 1: U-Net w/ condition is close in performance quantitatively and quantitatively, especially for AV45 dataset.
- Table 2: Impressive numbers for dataset AV45. The gap is not so large for AV1451, why?
- The statement "As Table 2 shows, for both the AV45 and AV1451, our method scores the highest in all metrics" is not correct. Some methods for AV1451 in Table 2 are better for some metrics, e.g. CLDM for Recall and Acc, Unet w/o condition and BPGAN both for AUC.
- Table 3: Why does performance with additional tabular data (PET* and Tabular) gets worse compared to PET*?
- The limitation of the additional metadata the model needs ($M$, $B_{CSF}$) to approximate $S$ and $B$ is not mentioned in the discussion. Did the comparison models also get the additional metadata? Do the presented experiments really analyze that the improved performance is coming from the causal model and not from the additional metadata available to the proposed approach?

**Detailed Comments:**

- There are inconsistency in the text: multimodal, multi-modal; also between the text and Figure 2: $\epsilon_{I_{PET}}$ and $\epsilon_{PET}$, $\epsilon_{I_{TI}}$ and $\epsilon_{TI}$.
- Figure 2: typos "Attntion" and "Fushion"; in general the figure and its font is too small.
- The discriminator loss (9) was not added to the overall optimization problem in (11). How is the discriminator trained? Minimax optimization?
- In my opinion it is confusing that the authors talk about decoders in Figure 2 but about generators in the equations.
- Would it also be possible to include prior information for $\epsilon_{PET}$ and $\epsilon_{TI}$; in this work they sample the prior information from the normal distribution.

**Justification Of The Final Rating:**

I thank the authors for the discussion, for fixing all the inconsistencies and typos, and for addressing my concerns. Taking everything into account, my final rating remains "weak accept" as I think this paper is worth discussing at the MIDL conference.

**Justification Of The Preliminary Rating:**

This work is a valuable addition to the literature on casual machine learning applied to medical images. The presented experiments are promising, but I think it still needs some cleanup and clarification before it can be finally accepted.

**Questions To Address In The Rebuttal:**

- Please fix the inconsistencies and typos mentioned in the detailed comments.
- Please answer all my questions and address my concerns from the weaknesses section.

**Special Issue:**

No

---

> ### Author Response · Authors · 2025-03-07
> **We have performed statistics test to validate the proposed method and corrected the typos.**
>
> 1. U-Net w/ condition achieves similar performance to our model in terms of MAE, PSNR, and SSIM on the AV45 dataset. **However, it underperforms in $\Epsilon_{SUVR}$ and downstream classification tasks, which suggests that our method better captures pathology-related characteristics rather than merely optimizing for pixel-level similarity.** This highlights the advantage of incorporating causal modeling to ensure that the generated images retain clinically meaningful features beyond surface-level reconstruction quality.  **To further support our claims, we have performed statistical significance testing (paired t-tests) to quantify improvements.**
>
> 2. **Since the overall pattern of the AV1451 dataset (the distribution of bright regions) is more distinct, as shown in Fig. 3, it is easier for the model to capture, resulting in higher synthesis quality.** Consequently, the performance gap between our method and others appears smaller. However, we conducted paired t-tests, which confirmed that our method still holds a statistically significant advantage over other methods.
>
> 3. We acknowledge that the statement "As shown in Table 2, our method achieves the highest scores in all metrics for both the AV45 and AV1451 datasets" is not entirely accurate. We have revised the statement accordingly.
>
> 4. Regarding the fusion of PET* and Tabular data, **our current implementation employs a relatively simple fusion mechanism, which may not fully leverage the complementary information from both modalities.** This could explain the performance drop when integrating tabular data. Future work could explore more sophisticated fusion techniques, such as learned feature alignment, to enhance the utilization of additional metadata. Moreover, retraining the downstream classifier with optimized data representations could potentially improve performance.
>
> 5. Not all comparison models utilize additional metadata. **Specifically, p2p and CycleGAN do not incorporate extra metadata, while the other baseline models do and the metadata utilization is the exactly same.** Given that our method operates under the same data constraints as those models, our performance gains can be attributed to the effectiveness of causal modeling, rather than merely benefiting from the availability of additional metadata. This reinforces the significance of explicitly modeling causal relationships in medical image synthesis to ensure that the generated images are both realistic and clinically meaningful.
>
> 6. We have corrected the overview figure accordingly (the typos, the font size and the “generator”).
>
> 7. Yes, the discriminator is trained using a standard adversarial framework, where the generator and discriminator are optimized in a minimax fashion. Specifically, the generator aims to minimize the adversarial loss while the discriminator maximizes it to distinguish between real and synthesized PET images. We have clarified this in the revised manuscript.
>
> 8. Incorporating prior information for $\epsilon_{I_{PET}}$ and $\epsilon_{I_{T1}}$ is indeed an interesting direction. In our current implementation, we assume a normal distribution as a flexible and generalizable prior, which aligns with common practices in generative modeling. However, we acknowledge that incorporating more informative priors—potentially derived from domain knowledge or empirical data—could further enhance the interpretability and accuracy of our model. We will explore this aspect in future work to assess its impact on the synthesis performance.

---

> ### Comment · Reviewer_gL8J · 2025-03-11
> **Open questions**
>
> I thank the authors for addressing my concerns and for the additional effort they put into the submission. However, I still have a few open points regarding the revision:
> - Table 2 and 3: Why have all the numbers changed? As I understood the authors' answer, they only did statistical significance tests. Compared to the previous version, are the new values now averages over ten different random seeds?
> - Additional metadata: Thank you for clarifying this point. I think the authors should add the information about which methods get additional metadata and which do not either in the tables or in the text.
> - Discriminator training: Unfortunately, I am still not satisfied with the overall loss function in (11). I would rather see explicitly how the minimax game was optimized, e.g:
> $$ L(AE, G_P, D) = L_{AE} + L_1(G_P) + \lambda_p L_{Perc.}(G_P) + \lambda_{adv}L(D)$$
> $$ (AE^*, G_P^*) = \arg \min_{AE, G_P} L(AE, G_P, D^*)$$
> $$ D^* = \arg \min_{D} - L(AE^*, G_P^*, D) $$
> And maybe rename $L(D)$ to $L_{GAN}$. Please correct me if I have presented your optimization problem incorrectly here.

---

> > ### Author Response · Authors · 2025-03-13
> >
> > 1. Yes, the new values now averages over ten different random seeds.
> >
> > 2. Thanks for the suggestions, we'll add the information about which methods get additional metadata in the text.
> >
> > 3. True, the optimization is exactly correct, thanks a lot!

---

### Official Review · Reviewer_kvrM · 2025-02-23

**Confidence:** 5
**Preliminary Rating:** 1

**Summary:**

This work generates synthetic PET images from T1 MRI images to be used for general interpretation by a physician

**Strengths:**

This work compares against many baseline works and performs replicates to estimate variance.
...............................................................................................................................

**Weaknesses:**

The idea of mapping from one modality to another for general interpretation is risky and will likely harm patients due to inevitable hallucinations (https://arxiv.org/abs/1805.08841)

The focus of the start of the paper is on generating synthetic images and then a lot of experiments focus on downstream tasks instead of studying the impact that the synthetic images can have. The paper should focus on one direction. The downstream classification task does not support that the synthetic images can improve clinical decision making.

The results are not significant. The stdev of the results is huge compared to the reported numbers. For example the MAE for AV45 shows "U-Net w/ condition" reports 0.237±0.142 while "Causal PETS" 0.224±0.104, these are essentially the same result, the difference in means is 10x smaller than the standard deviations! This also indicates that a simple unet can perform just as well as the complicated method proposed.

**Detailed Comments:**

.

**Justification Of The Preliminary Rating:**

As mentioned in the weaknesses the paper has multiple serious flaws.
....................................................................................................................................

**Questions To Address In The Rebuttal:**

Please provide evidence supporting that your approach is safe for patients.

**Special Issue:**

No

---

> ### Author Response · Authors · 2025-03-07
> **This reviewer has strong bias against cross-modality synthesis, which is unreasonable, as our work is backed by a hospital and Cross-modality image generation is not unexplored, numerous studies in this area have been developed with direct clinical support.**
>
> 1.**We strongly disagree with the suggestion that cross-modality mapping inherently leads to unreliable hallucinations. Our work is backed by a hospital (Xuanwu Hospital, Capital Medical University) and has been conducted in close collaboration with clinicians throughout the entire process. Additionally, we have already obtained preliminary clinical validation results, which further demonstrate the practical value of our method in real-world applications—these results will be detailed in a future publication. Cross-modality image generation is not an unexplored or inherently flawed approach; numerous studies in this area have been developed with direct clinical support, further legitimizing its viability** [1, 2, 3, 4]
>
> 2.As for the focus of our study, we emphasize that generating high-quality PET images is not our sole objective—our goal is to demonstrate their clinical relevance. **The downstream classification task is not a secondary consideration but rather a crucial validation step, as improved classification performance confirms that the synthetic PET images preserve critical pathological information.** To dismiss this as an unrelated task is to overlook a fundamental aspect of our study.
>
> 3.Regarding statistical significance, we firmly reject the claim that our results lack significance.** We have conducted rigorous statistical analyses, including paired t-tests, to confirm the superiority of our method over baseline approaches.** The improvements we report are not arbitrary; they are statistically validated and consistently observed across multiple metrics and datasets. While it is true that standard deviations are relatively large, this is an inherent characteristic of medical imaging due to subject variability, not a flaw in our approach. Importantly, our improvements remain consistent across evaluations, further reinforcing the robustness of our method.
>
> [1] Yi, Xin, Ekta Walia, and Paul Babyn. "Generative adversarial network in medical imaging: A review." Medical image analysis 58 (2019): 101552.
>
> [2] Guan, Hao, and Mingxia Liu. "Domain adaptation for medical image analysis: a survey." IEEE Transactions on Biomedical Engineering 69.3 (2021): 1173-1185.
>
> [3] Kong, Lingke, et al. "Breaking the dilemma of medical image-to-image translation." Advances in Neural Information Processing Systems 34 (2021): 1964-1978.
>
> [4] Park, E., Misra, S., Hwang, D.G. et al. Unsupervised inter-domain transformation for virtually stained high-resolution mid-infrared photoacoustic microscopy using explainable deep learning. Nat Commun 15, 10892 (2024). https://doi.org/10.1038/s41467-024-55262-2

---

> > ### Comment · Reviewer_kvrM · 2025-03-08
> >
> > > Our work is backed by a hospital
> >
> > The paper should contain support the claims of the paper, not the reputation of the authors or their institutions.
> >
> > > we have already obtained preliminary clinical validation results, which further demonstrate the practical value of our method in real-world applications—these results will be detailed in a future publication
> >
> > This does not satisfy my concerns with the current paper.
> >
> > > We have conducted rigorous statistical analyses, including paired t-tests, to confirm the superiority of our method over baseline approaches
> >
> > Where is this in the paper? I searched for the words significant, t-test, and p-value and I don't find anything.
> >
> > Can you share the analysis to the significance claims for your ablation study?

---

> > > ### Author Response · Authors · 2025-03-08
> > > **MIDL has strict page limits and primarily focuses on technical applications.**
> > >
> > > (1)  We have clinical experiments. We have also obtained preliminary clinical validation results, demonstrating the potential utility of our method in real-world applications. **However, MIDL has strict page limits and primarily focuses on technical applications. If we were to write the paper as you suggested, we would not be able to submit it to MIDL.**  This work was initiated by a hospital and has been conducted with continuous involvement from clinicians. The purpose of this project has never been to replace real MRI scans with generated images but rather to provide valuable insights from a causal perspective to assist doctors.
> > >
> > > (2)  Properly utilizing cross-modality image generation rather than outright rejection. Whether in natural or medical imaging, cross-modality image generation is a future trend. Instead of dismissing it outright, we should explore ways to use it correctly. Our lab has previously conducted two generation-based studies, both of which underwent direct clinical testing in hospitals. Additionally, arXiv papers are not peer-reviewed literature.
> > >
> > > (3)  arXiv papers lack authority and differ from our proposed idea. We strongly oppose using viewpoints from an arXiv paper to dismiss our three-year collaborative project with a top-tier hospital. Moreover, our approach strictly provides MRI synthesis as a reference for doctors, which is entirely different from the extreme validation approach used in that arXiv paper.
> > >
> > > (4)  The statistics test is provided in the **revised manuscripts (the Supporting Material in the rebuttal)**.

---

> > > > ### Comment · Reviewer_kvrM · 2025-03-08
> > > >
> > > > >  arXiv papers lack authority
> > > >
> > > > I shared the arxiv link to avoid a paywall. The paper was published in MICCAI: https://link.springer.com/chapter/10.1007/978-3-030-00928-1_60

---

> > > > ### Comment · Reviewer_kvrM · 2025-03-14
> > > >
> > > > > (4) The statistics test is provided in the revised manuscripts (the Supporting Material in the rebuttal).
> > > >
> > > > Thanks I see them now.
> > > >
> > > > In Table 1 I don't see how the significance is computed between the MAE "U-Net w/ condition" and "Causal PETS (ours)".
> > > > The difference  is labelled as ** (p < 0.01) but when I compute it for myself I find a different value.
> > > >
> > > > Do I have the number of samples wrong in this computation for AV45? In the appendix it states a "4:1:1 ratio" for data splits so 1/6 of the data is in the train set. If I add all the categories together I get (776+954+213)/6 = 323 which yields a p-value of 0.18.
> > > >
> > > > ```
> > > > scipy.stats.ttest_ind_from_stats(
> > > >     mean1=0.237,
> > > >     std1=0.142,
> > > >     nobs1=323,
> > > >     mean2=0.224,
> > > >     std2=0.104,
> > > >     nobs2=323,
> > > > )
> > > > pvalue=0.184
> > > > ```
> > > >
> > > > To me this experiment is very critical because if there is no difference here then it implies there is no difference between the proposed method and a conditioned U-Net.

---

> > > > > ### Author Response · Authors · 2025-03-14
> > > > >
> > > > > Thank you for your comment. The discrepancy in the p-value arises because we used a **paired t-test** (`scipy.stats.ttest_rel`), while the reviewer computed an **independent t-test** (`scipy.stats.ttest_ind_from_stats`).
> > > > >
> > > > > Since each subject has MAE values from both methods (`U-Net w/ condition` and `Causal PETS`), they are **paired measurements**, not independent samples. A **paired t-test** accounts for within-subject variability and provides greater statistical power than an independent t-test.
> > > > >
> > > > > In our analysis, we computed the difference between paired MAE values for each subject:
> > > > > \[
> > > > > D_i = X_i - Y_i
> > > > > \]
> > > > > where \(X_i\) and \(Y_i\) are the MAE values from `U-Net w/ condition` and `Causal PETS`, respectively. The paired t-test then assesses whether the mean difference is significantly different from zero.
> > > > >
> > > > > The reviewer’s independent t-test assumes that the two sets of MAE values are independent, ignoring the paired nature of the data. This results in a higher variance estimate, leading to a **larger p-value (0.184)** and a loss of statistical power.
> > > > >
> > > > > By using the appropriate paired t-test, we found **p < 0.01**, indicating a significant difference between the two methods. We believe this is the correct statistical approach given the paired nature of the data.

---

> > > > > > ### Comment · Reviewer_kvrM · 2025-03-14
> > > > > >
> > > > > > Can you share that data used to call scipy.stats.ttest_rel so I can compute the paired t-test? On the 7th I asked to "share the analysis to the significance claims for your ablation study" but have not received the analysis yet. In the paper I only see claims that the numbers are significant but not enough data to verify this. Perhaps this data can also be included in the appendix of the paper.

---

> > > > > > > ### Author Response · Authors · 2025-03-15
> > > > > > > **mae of CausalPETS  subset 1**
> > > > > > >
> > > > > > > It's hard and inappropriate to include the data in this chatbox(max characters 5000), however we tried best to meet the reviewer's requirement. We will add the significance claims for ablation study in the revised paper.
> > > > > > >
> > > > > > > The mae of CausalPETS  subset 1 [0.0213374339461326, 0.0219258686423301, 0.0161412910044193, 0.0211792914837598, 0.0232331407189369, 0.0229702618837356, 0.01798434073925, 0.018473573192954, 0.0162265962630509, 0.0206810587019511, 0.0223860303580759, 0.0230561335504055, 0.0185576406240463, 0.021661774483323, 0.0243855369210243, 0.0250526339679955, 0.0198003134250639, 0.0192922881364821, 0.0256663215279579, 0.0180413898944854, 0.0175675676882267, 0.0242963255316019, 0.023027195495367, 0.0238399234533309, 0.023740035381913, 0.0190971544206142, 0.0175756737589836, 0.0177249796688556, 0.0239453361302614, 0.0472524242102318, 0.0227047548562288, 0.0244333342581987, 0.0174363236874341, 0.0189857371151447, 0.0187643774718046, 0.0158977508544921, 0.024577193570137, 0.0185134565949439, 0.0147185748636722, 0.0346705216566176, 0.0306641870856284, 0.0308234059929846, 0.022805471804738, 0.0233920448392629, 0.0233559949815273, 0.0194314484417437, 0.0172124042630195, 0.0225484004735946, 0.0192030972361564, 0.0239234686762094, 0.0443771851181984, 0.0257640668869017, 0.0125152288973331, 0.028240062457323, 0.0185953583568334, 0.0234341499328613, 0.0143551807850599, 0.011922397211194, 0.013850687071681, 0.0145364254713058, 0.0207690046101808, 0.0255764645457267, 0.0230507896214723, 0.0100785529986023, 0.0238127155721186, 0.0207599647462368, 0.0185624491423368, 0.0505832770586012, 0.013903905314207, 0.0185872769474983, 0.0146190828204154, 0.0146614480763673, 0.0202986382812261, 0.0309642470747231, 0.0218921777725219, 0.0222519141554832, 0.0131869949400424, 0.0132171148434281, 0.0333491963267326, 0.0175154757618904, 0.021290863764286, 0.0225274999558925, 0.031578564953804, 0.0182583377629518, 0.0597870761275291, 0.0169972466796636, 0.0287198136866092, 0.020902503889799, 0.0436036804437637, 0.0171957351267337, 0.0166614037007093, 0.0288700446519848, 0.0204363138347864, 0.0216722235947846, 0.0247398325175045, 0.015591730363667, 0.0154956318438053, 0.0160586878156238, 0.0176672507077455, 0.0172097757458686, 0.016783483326435, 0.0226162475466728, 0.0126709364473818, 0.0191284473166557, 0.0248181623339653, 0.0543570855379104, 0.0217405249297618, 0.010641778082628, 0.0224256840467453, 0.0254412432134151, 0.0184147047370672, 0.0263776321828365, 0.0019247123393606, 0.0163943096995353, 0.0164758190512657, 0.0138833867385983, 0.0140245063230395, 0.0197125085562466, 0.0179260344177484, 0.0184703738361597, 0.0264148250341414, 0.0206311760962009, 0.0190703831613063, 0.0188781842589378, 0.0253330214440821, 0.0177898785591124, 0.0267581482350825, 0.0270161432147026, 0.0246418760091066, 0.0417028426528298, 0.02201484785676, 0.0154633857309818, 0.0145781524479389, 0.014910889789462, 0.0211661205559968, 0.0198311888545751, 0.0253069941624813, 0.019623951986432, 0.0184659101068973, 0.0190786953974788, 0.019665780377388, 0.0253733833253383, 0.0176269103705883, 0.021523109537363, 0.0173515927910803, 0.0175153882175683, 0.018204443988204, 0.0194611549377441, 0.0173650942742824, 0.0209249226868151, 0.0634365043164422, 0.0257415425896644, 0.0212026079118251, 0.019484559905529, 0.0214501439154148, 0.0189399001121521, 0.0526082074761389, 0.0180181962370872, 0.0183107488721609, 0.016008099541068, 0.0155715877190232, 0.01673393137753, 0.0254088641822337, 0.0133286124587058, 0.0155590577051043, 0.016514353454113, 0.0183908966302871, 0.0196265083670616, 0.0177528624355793, 0.0195701960474252, 0.022599362668395, 0.0236978122711181, 0.0177238887667655, 0.0200573884814976, 0.0194018745541571, 0.0181690330177544, 0.0191638004034757, 0.0100395238518714, 0.0177939337611198, 0.0184049382805824, 0.0195331991583108, 0.0175535482287405, 0.0311039473235607, 0.025108688187599, 0.0182738682746886, 0.0196140062928199, 0.0212854397416114, 0.0225489296883344, 0.0213498034447431, 0.019933545961976, 0.0128779802471399, 0.0136684393510222, 0.0423470431685447, 0.0060187203437089, 0.0088540170875191, 0.0271749877303838, 0.0270190079629421, 0.0558440575122832, 0.0178066854298114, 0.0177767556160688]

---

> > > > > > > > ### Author Response · Authors · 2025-03-15
> > > > > > > > **mae of CausalPETS  subset 2**
> > > > > > > >
> > > > > > > > [0.0307200989991426, 0.0185561310499906, 0.016790946945548, 0.0304026707381009, 0.0173857313513755, 0.0051089616119861, 0.0058623401075601, 0.017494574189186, 0.0194402773797511, 0.0223766517758369, 0.041889894169569, 0.0119190076773197, 0.0182173270732164, 0.0194230005145072, 0.0609347784638404, 0.017386336711049, 0.0174182305604219, 0.0256432675689458, 0.0238051571816205, 0.0208058379709719, 0.0197001312792301, 0.0406709437936544, 0.0201484438896179, 0.0190367129564285, 0.0215051778525114, 0.0237213456004858, 0.0185836050659418, 0.0068884582817554, 0.0186283668994903, 0.026592821714282, 0.016673756763339, 0.0176731952399015, 0.0247875087827444, 0.0206606079757213, 0.0207918574035167, 0.0188138996183872, 0.0184033130228519, 0.0210353721797466, 0.0205617946505546, 0.043993242151407, 0.0125804794952273, 0.0128897773101925, 0.0143923456361307, 0.0162603929638862, 0.019414551270008, 0.019045565277338, 0.0181861565381288, 0.0090597139298915, 0.0121294543266295, 0.0119482301294802, 0.0237622842311859, 0.0207342644155025, 0.0180071693778038, 0.0262104055404663, 0.1093348547935485, 0.0193735946834087, 0.0129827536537928, 0.0178484376519918, 0.0164800379425287, 0.0155946696177124, 0.0187387466430664, 0.0238604702919721, 0.0182978684574364, 0.0470923671245574, 0.0204818759977817, 0.0210265302032232, 0.0252901882797479, 0.0265274484574794, 0.0249780815958976, 0.0174493067085741, 0.0193208594739437, 0.0184037619203329, 0.0191135973602533, 0.0175450471162795, 0.0257063050687313, 0.0197456783175468, 0.1089594124165058, 0.0239339777201414, 0.0183603045463562, 0.0298084645837545, 0.0183976335942744, 0.0120989316849651, 0.0178666532903909, 0.0193181429058313, 0.0199463404715061, 0.0195727822303771, 0.0226343636333942, 0.0333082193255423, 0.0230603636175394, 0.0190159295618533, 0.0260660358637571, 0.0210330960273742, 0.0259117005884647, 0.0176026326537131, 0.0188427194952964, 0.0220327944427728, 0.0195399709045887, 0.0441417136907576, 0.016923539340496, 0.0187883124619721, 0.0216555249810218, 0.0241346926361321, 0.0269374290227889, 0.0222326970219611, 0.0253161274760961, 0.0206827771782875, 0.0175088987618684, 0.0172559712082147, 0.016515590250492, 0.0165687371045351, 0.0181627497076988, 0.0191965345293283, 0.019233067420125, 0.0313474872827528, 0.0249018044143915, 0.0254897390425205, 0.0194688410431146, 0.0189554765820503, 0.0295060204088687, 0.0187132656574249, 0.0182476405918598, 0.0233324141174554, 0.0185977388173341, 0.018012951028347, 0.0210693099615367, 0.0268735610067843, 0.0192623831450938, 0.0246272871017456, 0.0264490381002426, 0.0241205267727375, 0.017417815414071, 0.0195269621908664, 0.0570829326033591, 0.0269630482912062, 0.0209952396273612, 0.0181143292158841, 0.0194555808722972, 0.015389195565279, 0.0384389349937438, 0.0401602053761481, 0.0164814352512359, 0.0166707602560519, 0.03061814994812, 0.013849850744009, 0.0399569684386252, 0.0188100803643465, 0.0289977719306944, 0.0260885572701692, 0.0363561922430991, 0.022113957118988, 0.0244383925318716, 0.0252143092572688, 0.0243185296773909, 0.0225627742826938, 0.0223524534821509, 0.0257442993044853, 0.0190718366324901, 0.0323091188192367, 0.0232525991022585, 0.023788128209114, 0.0237905049443244, 0.0231918566048145, 0.0263573251783846, 0.0284041428685187, 0.0172699244439601, 0.0271260951578617, 0.0226734005033968, 0.0229952784001827]

---

> > > > > > > > > ### Author Response · Authors · 2025-03-15
> > > > > > > > > **mae of unet* subset 1**
> > > > > > > > >
> > > > > > > > > [0.0213374339461326, 0.0219258686423301, 0.0161412910044193, 0.0211792914837598, 0.0232331407189369, 0.0229702618837356, 0.01798434073925, 0.018473573192954, 0.0162265962630509, 0.0206810587019511, 0.0223860303580759, 0.0230561335504055, 0.0185576406240463, 0.021661774483323, 0.0243855369210243, 0.0250526339679955, 0.0198003134250639, 0.0192922881364821, 0.0256663215279579, 0.0180413898944854, 0.0175675676882267, 0.0242963255316019, 0.023027195495367, 0.0238399234533309, 0.023740035381913, 0.0190971544206142, 0.0175756737589836, 0.0177249796688556, 0.0239453361302614, 0.0472524242102318, 0.0227047548562288, 0.0244333342581987, 0.0174363236874341, 0.0189857371151447, 0.0187643774718046, 0.0158977508544921, 0.024577193570137, 0.0185134565949439, 0.0147185748636722, 0.0346705216566176, 0.0306641870856284, 0.0308234059929846, 0.022805471804738, 0.0233920448392629, 0.0233559949815273, 0.0194314484417437, 0.0172124042630195, 0.0225484004735946, 0.0192030972361564, 0.0239234686762094, 0.0443771851181984, 0.0257640668869017, 0.0125152288973331, 0.028240062457323, 0.0185953583568334, 0.0234341499328613, 0.0143551807850599, 0.011922397211194, 0.013850687071681, 0.0145364254713058, 0.0207690046101808, 0.0255764645457267, 0.0230507896214723, 0.0100785529986023, 0.0238127155721186, 0.0207599647462368, 0.0185624491423368, 0.0505832770586012, 0.013903905314207, 0.0185872769474983, 0.0146190828204154, 0.0146614480763673, 0.0202986382812261, 0.0309642470747231, 0.0218921777725219, 0.0222519141554832, 0.0131869949400424, 0.0132171148434281, 0.0333491963267326, 0.0175154757618904, 0.021290863764286, 0.0225274999558925, 0.031578564953804, 0.0182583377629518, 0.0597870761275291, 0.0169972466796636, 0.0287198136866092, 0.020902503889799, 0.0436036804437637, 0.0171957351267337, 0.0166614037007093, 0.0288700446519848, 0.0204363138347864, 0.0216722235947846, 0.0247398325175045, 0.015591730363667, 0.0154956318438053, 0.0160586878156238, 0.0176672507077455, 0.0172097757458686, 0.016783483326435, 0.0226162475466728, 0.0126709364473818, 0.0191284473166557, 0.0248181623339653, 0.0543570855379104, 0.0217405249297618, 0.010641778082628, 0.0224256840467453, 0.0254412432134151, 0.0184147047370672, 0.0263776321828365, 0.0019247123393606, 0.0163943096995353, 0.0164758190512657, 0.0138833867385983, 0.0140245063230395, 0.0197125085562466, 0.0179260344177484, 0.0184703738361597, 0.0264148250341414, 0.0206311760962009, 0.0190703831613063, 0.0188781842589378, 0.0253330214440821, 0.0177898785591124, 0.0267581482350825, 0.0270161432147026, 0.0246418760091066, 0.0417028426528298, 0.02201484785676, 0.0154633857309818, 0.0145781524479389, 0.014910889789462, 0.0211661205559968, 0.0198311888545751, 0.0253069941624813, 0.019623951986432, 0.0184659101068973, 0.0190786953974788, 0.019665780377388, 0.0253733833253383, 0.0176269103705883, 0.021523109537363, 0.0173515927910803, 0.0175153882175683, 0.018204443988204, 0.0194611549377441, 0.0173650942742824, 0.0209249226868151, 0.0634365043164422, 0.0257415425896644, 0.0212026079118251, 0.019484559905529, 0.0214501439154148, 0.0189399001121521, 0.0526082074761389, 0.0180181962370872, 0.0183107488721609, 0.016008099541068, 0.0155715877190232, 0.01673393137753, 0.0254088641822337, 0.0133286124587058, 0.0155590577051043, 0.016514353454113, 0.0183908966302871, 0.0196265083670616, 0.0177528624355793, 0.0195701960474252, 0.022599362668395, 0.0236978122711181, 0.0177238887667655, 0.0200573884814976, 0.0194018745541571, 0.0181690330177544, 0.0191638004034757, 0.0100395238518714, 0.0177939337611198, 0.0184049382805824, 0.0195331991583108, 0.0175535482287405, 0.0311039473235607, 0.025108688187599, 0.0182738682746886, 0.0196140062928199, 0.0212854397416114, 0.0225489296883344, 0.0213498034447431, 0.019933545961976, 0.0128779802471399, 0.0136684393510222, 0.0423470431685447, 0.0060187203437089, 0.0088540170875191, 0.0271749877303838, 0.0270190079629421, 0.0558440575122832, 0.0178066854298114, 0.0177767556160688]

---

> > > > > > > > > > ### Author Response · Authors · 2025-03-15
> > > > > > > > > > **mae of unet* subset 2**
> > > > > > > > > >
> > > > > > > > > > [0.0307200989991426, 0.0185561310499906, 0.016790946945548, 0.0304026707381009, 0.0173857313513755, 0.0051089616119861, 0.0058623401075601, 0.017494574189186, 0.0194402773797511, 0.0223766517758369, 0.041889894169569, 0.0119190076773197, 0.0182173270732164, 0.0194230005145072, 0.0609347784638404, 0.017386336711049, 0.0174182305604219, 0.0256432675689458, 0.0238051571816205, 0.0208058379709719, 0.0197001312792301, 0.0406709437936544, 0.0201484438896179, 0.0190367129564285, 0.0215051778525114, 0.0237213456004858, 0.0185836050659418, 0.0068884582817554, 0.0186283668994903, 0.026592821714282, 0.016673756763339, 0.0176731952399015, 0.0247875087827444, 0.0206606079757213, 0.0207918574035167, 0.0188138996183872, 0.0184033130228519, 0.0210353721797466, 0.0205617946505546, 0.043993242151407, 0.0125804794952273, 0.0128897773101925, 0.0143923456361307, 0.0162603929638862, 0.019414551270008, 0.019045565277338, 0.0181861565381288, 0.0090597139298915, 0.0121294543266295, 0.0119482301294802, 0.0237622842311859, 0.0207342644155025, 0.0180071693778038, 0.0262104055404663, 0.1093348547935485, 0.0193735946834087, 0.0129827536537928, 0.0178484376519918, 0.0164800379425287, 0.0155946696177124, 0.0187387466430664, 0.0238604702919721, 0.0182978684574364, 0.0470923671245574, 0.0204818759977817, 0.0210265302032232, 0.0252901882797479, 0.0265274484574794, 0.0249780815958976, 0.0174493067085741, 0.0193208594739437, 0.0184037619203329, 0.0191135973602533, 0.0175450471162795, 0.0257063050687313, 0.0197456783175468, 0.1089594124165058, 0.0239339777201414, 0.0183603045463562, 0.0298084645837545, 0.0183976335942744, 0.0120989316849651, 0.0178666532903909, 0.0193181429058313, 0.0199463404715061, 0.0195727822303771, 0.0226343636333942, 0.0333082193255423, 0.0230603636175394, 0.0190159295618533, 0.0260660358637571, 0.0210330960273742, 0.0259117005884647, 0.0176026326537131, 0.0188427194952964, 0.0220327944427728, 0.0195399709045887, 0.0441417136907576, 0.016923539340496, 0.0187883124619721, 0.0216555249810218, 0.0241346926361321, 0.0269374290227889, 0.0222326970219611, 0.0253161274760961, 0.0206827771782875, 0.0175088987618684, 0.0172559712082147, 0.016515590250492, 0.0165687371045351, 0.0181627497076988, 0.0191965345293283, 0.019233067420125, 0.0313474872827528, 0.0249018044143915, 0.0254897390425205, 0.0194688410431146, 0.0189554765820503, 0.0295060204088687, 0.0187132656574249, 0.0182476405918598, 0.0233324141174554, 0.0185977388173341, 0.018012951028347, 0.0210693099615367, 0.0268735610067843, 0.0192623831450938, 0.0246272871017456, 0.0264490381002426, 0.0241205267727375, 0.017417815414071, 0.0195269621908664, 0.0570829326033591, 0.0269630482912062, 0.0209952396273612, 0.0181143292158841, 0.0194555808722972, 0.015389195565279, 0.0384389349937438, 0.0401602053761481, 0.0164814352512359, 0.0166707602560519, 0.03061814994812, 0.013849850744009, 0.0399569684386252, 0.0188100803643465, 0.0289977719306944, 0.0260885572701692, 0.0363561922430991, 0.022113957118988, 0.0244383925318716, 0.0252143092572688, 0.0243185296773909, 0.0225627742826938, 0.0223524534821509, 0.0257442993044853, 0.0190718366324901, 0.0323091188192367, 0.0232525991022585, 0.023788128209114, 0.0237905049443244, 0.0231918566048145, 0.0263573251783846, 0.0284041428685187, 0.0172699244439601, 0.0271260951578617, 0.0226734005033968, 0.0229952784001827]

---

> > > > > > > > > > > ### Author Response · Authors · 2025-03-15
> > > > > > > > > > >
> > > > > > > > > > > The reviewer calculated the number of test scans as (776+954+213)/6 = 323, assuming an even split of scans among six folds. However, our dataset is split at the **subject level** rather than the **scan level** to prevent data leakage. Since each subject can have a varying number of scans, the actual number of scans in the test set does not necessarily match the proportion implied by a direct scan-wise division. Instead, we randomly assign subjects to the 4:1:1 split, which naturally results in variations in the number of scans per set.

---

> ### Author Response · Authors · 2025-03-08
>
> 1. **Many subsequent studies have further advanced image translation techniques in MIDL, MICCAI, TMI and MIA since 2018**. In MIDL 2024 alone, five papers[1-5] on this topic were published, with another five in MICCAI 2024 and MICCAI 2023, along with numerous others in top-tier journals such as MIA and TMI[6-10]. Following the logic of the paper in 2018, all these works should have been rejected as well.
>
> 2. In addition, the idea of our work is different from the paper in 2018. This project was never intended to substitute real PET scans with generated images. Instead, its goal is to offer valuable insights through a causal framework to support medical decision-making. Clinicians can use our approach as a reference to refine their diagnoses, helping to determine the most appropriate treatments and dosages for patients at different stages.
>
> [1] Ji, Yanni, Marie Cutiongco, and Ke Yuan. "CP2Image: Generating high-quality single-cell images using CellProfiler representations." Medical Imaging with Deep Learning. PMLR, 2024.
>
> [2] Omidi, Abbas, et al. "Unsupervised domain adaptation of brain MRI skull stripping trained on adult data to newborns: Combining synthetic data with domain invariant features." Medical Imaging with Deep Learning. PMLR, 2024.
>
> [3] Dey, Avirup, and Mehran Ebrahimi. "Mtsr-mri: Combined modality translation and super-resolution of magnetic resonance images." Medical Imaging with Deep Learning. PMLR, 2024.
>
> [4] Xin, Bowen, et al. "Deformation-aware GAN for Medical Image Synthesis with Substantially Misaligned Pairs." Medical Imaging with Deep Learning. PMLR, 2024.
>
> Saeed, Shaheer U., et al. "Bi-parametric prostate MR image synthesis using pathology and sequence-conditioned stable diffusion." Medical imaging with deep learning. PMLR, 2024.
>
> [5] Zingman, Igor, et al. "A comparative evaluation of image-to-image translation methods for stain transfer in histopathology." Medical Imaging with Deep Learning. PMLR, 2024.
>
> [6] Özbey, Muzaffer, et al. "Unsupervised medical image translation with adversarial diffusion models." IEEE Transactions on Medical Imaging 42.12 (2023): 3524-3539.
>
> [7] Wang, Clinton J., Natalia S. Rost, and Polina Golland. "Spatial-intensity transforms for medical image-to-image translation." IEEE transactions on medical imaging 42.11 (2023): 3362-3373.
>
> [8] Li, Yunxiang, et al. "Zero-shot medical image translation via frequency-guided diffusion models." IEEE transactions on medical imaging 43.3 (2023): 980-993.
>
> [9] Zhou, Yinchi, et al. "Cascaded multi-path shortcut diffusion model for medical image translation." Medical Image Analysis 98 (2024): 103300.
>
> [10] Wang, Zihao, et al. "Mutual information guided diffusion for zero-shot cross-modality medical image translation." IEEE Transactions on Medical Imaging (2024).

---

> > ### Comment · Reviewer_kvrM · 2025-03-14
> >
> > > This project was never intended to substitute real PET scans with generated images
> >
> > You write in the paper "and the health risks posed by radiation exposure. Thus, there is an urgent need to explore alternative approaches for acquiring PET to support diagnostic applications, among which the synthesis of PET from other more available modalities presents a promising solution." which sounds like substituting real PET scans with generated synthetic ones.

---

### Official Review · Reviewer_mFuE · 2025-02-23

**Confidence:** 4
**Preliminary Rating:** 3
**Final Rating:** 4

**Summary:**

This paper proposes a novel causality-informed model for synthesizing PET images from multi-modal data, including T1-weighted MRI, cerebrospinal fluid (CSF) biomarkers, and demographic factors, addressing the high cost and radiation risks of traditional PET scans. Unlike conventional deep learning models, it explicitly models causal relationships using causal graphs and structural causal equations (SCEs) to improve synthesis accuracy and interpretability. The model incorporates an auto-encoder, an attention-based PET generator, and a GAN-based discriminator, trained with ADNI dataset for Alzheimer’s Disease (AD) diagnosis. Experiments demonstrate that Causal PETS outperforms existing methods in PET image synthesis and improves disease classification (pMCI vs. sMCI).

**Strengths:**

1. Unlike conventional deep learning-based PET synthesis methods, Causal PETS explicitly models causal relationships among multimodal inputs (MRI, CSF biomarkers, demographics), improving interpretability and robustness.
2. The use of causal graphs and structural causal equations (SCEs) ensures that the model accounts for underlying biological mechanisms, rather than relying solely on data-driven correlations.
3. Causal PETS effectively fuses complementary information from T1 MRI, CSF biomarkers, and demographic factors, overcoming the information gap between different imaging modalities.

**Weaknesses:**

1. This work lacks a discussion on important prior research in causal modeling, particularly Deep Structural Causal Models (DSCMs) (2020) and Neural Causal Models (NCMs) (2022), along with their follow-up studies e.g. De Sousa Ribeiro et al 2023. These works have explored counterfactual generation within the Structural Causal Model (SCM) framework, adhering to Pearl's causality ladder, and explicitly incorporating abduction, intervention, and counterfactual steps. I believe a discussion with these previous works would be helpful.
2. Another limitation of this work is the absence of an abduction step, which is a crucial component in counterfactual reasoning as outlined in Pearl’s causality framework. Abduction involves estimating the exogenous noise variables that encode individual-specific information, allowing for more precise counterfactual predictions while preserving subject identity in the generated images. Since Causal PETS does not explicitly perform abduction, it is important to discuss how the method ensures that the synthesized PET images retain the unique characteristics of the original subject rather than producing generic or overly smoothed representations.
3. The causal graph (Figure 1) plays a central role in the proposed method, yet its construction is not well justified. It is unclear whether the causal relationships were derived from clinical evidence, expert domain knowledge, or data-driven discovery.
4. What is the size of image in this paper? I found figure 3 very blurry and hard to interpret  the quality of generated images.

**Detailed Comments:**

See weakness.

**Justification Of The Final Rating:**

After further consideration, I have decided to increase my score to weak acceptance. Upon reflection, I realize that the blurriness of the PET images in grayscale affected my initial judgment. If possible, changing them to a heatmap representation would enhance clarity.

**Justification Of The Preliminary Rating:**

This paper introduces a causality-informed PET synthesis framework that improves image generation by modeling causal relationships among multimodal data. The approach is promising, with strong experimental results and a counterfactual analysis component that enhances interpretability.

However, significant weaknesses limit its impact. The paper fails to engage with key causal generative models (e.g., DSCMs 2020, NCMs 2022), lacks abduction, and does not validate its causal graph. As such, I vote for borderline.

**Questions To Address In The Rebuttal:**

See weakness.

---

> ### Author Response · Authors · 2025-03-07
> **We have added the discussion of DSCMs and NCMs, we explained why we don't include abduction step and how to retain subject identity because:**
>
> 1. We acknowledge the Deep Structural Causal Models (DSCMs, 2020) and Neural Causal Models (NCMs, 2022); however, we did not explicitly compare our work with them because our fundamental ideas are aligned. In fact, our work can be seen as a further extension of DSCMs:
> (1) Expanded to More Data Modalities. Unlike DSCMs, which primarily focused on a single data modality, our approach extends the framework to incorporate multiple modalities.
> (2) Improved Image Generation. While the DSCM paper employed VAE for image generation, we utilize a more advanced generator (GAN), leading to improved quality and realism in synthesized images.
> (3) Defined in a Verifiable Setting. Our method is designed within a ground-truth-verifiable setting, allowing for objective validation of causal generation. In contrast, the counterfactual images produced by DSCMs (2020) lacked ground truth for validation making their effectiveness harder to assess.
> **We have included a discussion on this in our revised paper, please refer to introduction section.**
>
> 2. Our paper does not include an abduction step because **our main task, synthesizing PET from T1 MRI does not involve counterfactual prediction**, which distinguishes our setting from DSCMs (2020).
> In our causal graph, $S$ represents structural information of the brain, while $B$ encodes the distribution of pathological molecules. Estimating exogenous noise variables only accounts for a small portion of PET image noise. As shown in our ablation study, removing the exogenous noise variables does not significantly impact model performance. Therefore, when generating PET images based on the causal graph, we sample from a normal distribution to simulate PET imaging noise instead.
> Our approach ensures that the synthesized PET images retain subject identity because:
> (1) $S$ provides structural information, and $B$ provides pathological molecular distribution, offering sufficient information for identity preservation.
> (2) Pixel-wise L1 loss (with ground truth) and adversarial loss from the discriminator further constrain the model, guiding it to generate subject-specific PET images rather than generic or overly smoothed ones.
> The classification results of sMCI and pMCI in our downstream task further confirm the clinical relevance and subject specificity of our synthesized PET images, demonstrating that they are not merely generic or overly smoothed representations.
> 3. The construction of the causal graph is based on clinical evidence and expert domain knowledge. Specifically, the methodology for establishing each edge is described in Sec 2.1.1.
> 4. The image size used in this study is 96 × 128 × 96, with a resolution of 1.5 mm. The blurriness is an inherent characteristic of PET images due to imaging principle: PET relies on the metabolic distribution of a radioactive tracer (e.g., 18F-FDG). The scanner detects gamma-ray pairs emitted from positron annihilation. However, before annihilation, positrons travel a short distance in tissue (positron range effect), reducing spatial resolution and introducing blurring.

---

> ### Comment · Reviewer_mFuE · 2025-03-08
>
> - I do not agree that this work can be seen as an extension of DSCMs. The ways they implemented causal equations/mechanisms are different. While DSCMs 2020, 2023 enabled invertible causal mechanisms, this work uses some neural works to take e.g. A, B as input, and outputs C, claiming A, B are parents of C. This is not invertible. I think this work and DSCM are different types of methods.
> - DSCM 2020 used a VAE, but De Sousa Ribeiro et al 2023 improved DSCM with a HVAE for high resolution image synthesis, which I also mentioned in the original review. Also, claiming GAN is a more advanced generator over VAE is not a selling point to compare with a paper from 2020, as "more advanced generator" in this age should be diffusion models.
> - My concern about construction of causal graph is solved.
>
> My score remains at borderline but I will lower my confidence score as I can't interpret if the synthesised PET images are good or not.

---

> > ### Author Response · Authors · 2025-03-09
> >
> > 1. **We agree with the reviewer that our approach is different from DSCMs; rather than being a direct extension, our work is motivated by theirs.** However, while DSCMs implement invertible causal mechanisms, their image generation process does not achieve true invertibility. Instead, they use a decoder to approximate the inverse process. Similarly, if needed, we could also employ a decoder to model this inverse process. However, since our focus is not on generating counterfactual images, we did not emphasize this aspect in our work.
> >
> > 2. **We acknowledge that the advancement of GANs is not our primary selling point. Our goal is to establish a causal image translation framework that allows flexibility in choosing different generative models.** In our current dataset, GANs are the most suitable choice. Regarding De Sousa Ribeiro et al. (2023), their work remains limited to 2D, and their network architecture is too large to be applied to 3D image synthesis as we have done. We attempted to reproduce their model on 24GB GPU, but even with a batch size of 1, it was not feasible. Additionally, we also experimented with diffusion models. However, given the scale of our dataset, diffusion models performed poorly, which led us to abandon the diffusion-based framework.

---

> > > ### Comment · Reviewer_mFuE · 2025-03-09
> > >
> > > Thank the authors for the response.
> > > 1. For image variables, DSCM's causal mechanism is not truly invertible. But for non-image lower-dimensional variables, their causal mechanisms are truly invertible with the use of normalizing flows.
> > > 2. This detailed information is helpful. What is the size of the dataset? As reported in Table 5, there seems to be only hundreds of images for training. Since the authors used ADNI, I remember ADNI should contain tens of thousands of images, why not use more?

---

> > > > ### Comment · Reviewer_mFuE · 2025-03-10
> > > >
> > > > Another thing I want to suggest is maybe instead of gray scale image, it would be better to show the PET results of Fig. 3 in color form as heatmap.

---

> > > > > ### Author Response · Authors · 2025-03-13
> > > > >
> > > > > Thanks for the suggestions, we'll change to heatmap.
> > > > >
> > > > > The limited number of data is due to the required multi-modality.

---

> > > > > > ### Comment · Reviewer_mFuE · 2025-03-13
> > > > > >
> > > > > > Thank authors for response. After further consideration, I have decided to increase my score to weak acceptance. Upon reflection, I realize that the blurriness of the PET images in grayscale affected my initial judgment. If possible, changing them to a heatmap representation would enhance clarity.

---

### Author Rebuttal · Authors · 2025-03-07

**Rebuttal:**

We submitted a revised manusript, with corrected typos, more analysis, added statistics test and more experiments.

(1) The reviewer mFuE mainly concerns about the limited discussion on important prior research in causal modeling and the difference of our work compared with the prior research. We have added the discussion in the revised manusript.

(2) Reviewer kvrM has strong bias against cross-modality synthesis, which is unreasonable. We explained that our work is backed by a hospital. We cited the related literature to prove that cross-modality image generation is not unexplored and numerous studies in this area have been developed with direct clinical support.

(3) Reviewer gL8J concerns about the performance of our proposed model, which can be validated by the added statistics test. We have corrected the mentioned typos and answered all the questions.

(4) Reviewer xGdK questioned about the inappropriate input for clinical scenarios, limited novelty and the missing meta-ROI-based analysis. We explained that our previous work on **arxiv** wasn't and will not be published in any peer-reviewed conference or journal, which is primarily served as an initial exploration. Therefore, it should not be used as a definitive benchmark to diminish the contributions of our work. We also explained the clinical significance and added meta-ROI-based analysis.

**Supporting Material:**

/attachment/5dd5d88e2b48f10cc9cf5202ded5cc934afc2887.pdf

---

### Comment · Area_Chair_22vb · 2025-03-10
**Etiquette**

This is a kindly reminder to both authors and the reviewers to engage in polite and fruitful discussion. This is clearly mentioned in the code-of-conduct of MIDL (https://2025.midl.io/code-of-conduct). Kindly refrain from becoming too aggressive during the discussion period.

Thanks,
Area Chair handling this paper

---

> ### Author Response · Authors · 2025-03-13
>
> Thank you for your reminder, and we apologize if this has caused any inconvenience. We appreciate the straightforward feedback on the quality issues. However, we are deeply disheartened that some comments are overly aggressive, and we can't revise our work accordingly.
>
> As pointed out by Reviewer kvrM and Reviewer xGdK: "The image generation for medical purposes is meaningless," even though many papers focus on this area. Additionally, MIDL does not recognize a specific dataset, even though it is considered a benchmark in the field. They further provided suggestions for alternative datasets, but they themselves acknowledged that these datasets cannot validate our method.
>
> These decisions appear to stem from their individual perspectives rather than a consensus within the broader community, and it is unfeasible for us to revise our work to meet their standards. In other words, our four years of effort are rendered entirely meaningless.

---

### Meta-Review · Area_Chair_22vb · 2025-03-16

**Recommendation:** Accept (Poster)
**Confidence:** 4

**Metareview:**

The paper proposes a causality-informed PET image synthesis from multi-modal data. Initial reviews were quite polarised; however, by the end of the discussion period, final recommendations were three weak accept and one strong reject.

Before the rebuttal period, the first reviewer was not particularly happy about the causality link and blurriness of the generated images; however, after the rebuttal period, the authors provided better references for causality, and the reviewer acknowledged their shortcomings in evaluating generated PET images (which are in general blurry compared to T1). I encourage authors to change generated grayscale PET images to color-coded images for the camera-ready version.

The second reviewer was particularly harsh on the paper and recommended strong rejection. However, looking into the discussion, I can clearly see that the reviewer was biased against methods that use GAN for image generation. Additionally, the reviewer asked the author to upload all values of the experimental study so that they could verify the generated p-values. This is particularly harsh for the authors and shows the distrust and bias of the reviewer. The MIDL community highly discourages such behavior during the review period. Considering all this, I am ignoring the recommendation of this reviewer. Also, I would like to apologize to the authors for any inconvenience this caused during the discussion period.

The third reviewer clearly saw the utility of the proposed work and recommended weak acceptance (both before and after rebuttal).

The last reviewer initially recommended weak reject and, during the discussion period, went to strong reject. Still, in the end, once the authors performed the experiments recommended by the reviewer, they recommended weak accept. Initial weak reject (and subsequent strong reject) was based on the utility of the proposed method in clinical settings and smaller dataset sizes. This was finally resolved at the end of the discussion period. Again, I would like to apologize to the authors for the heated discussion with this reviewer. The MIDL community doesn't recommend this type of behavior.

Overall, considering everything, I am recommending acceptance of the paper.